**communications** engineering

# An integrated coupled oscillator network to solve optimization problems
**Markus Graber** ✉ & **Klaus Hofmann**

Solving combinatorial optimization problems is essential in scientific, technological, and engineering applications, but can be very time and energy-consuming using classical algorithms executed on digital processors. Oscillator-based Ising machines offer a promising alternative by exploiting the analog coupling between electrical oscillators to solve such optimization problems more efficiently. Here we present the design and the capabilities of our scalable approach to solve Ising and quadratic unconstrained binary optimization problems. This approach includes routable oscillator connections to simplify the time-consuming embedding of the problem into the oscillator network. Our manufactured silicon chip, featuring 1440 oscillators implemented in a 28 nm technology, demonstrates the ability to solve optimization problems in 950 ns while consuming typically 319 µW per node. A frequency, phase, and delay calibration ensures robustness against manufacturing variations. The system is evaluated with multiple sets of benchmark problems to analyze the sensitivity for parameters such as the coupling strength or frequency.

The scaling of semiconductor technologies to smaller, more advanced nodes to increase computing power and energy efficiency is getting more and more difficult with each semiconductor technology step. In addition, the demand for computing power in combination with improved energy efficiency is increasing, particularly due to advances in artificial intelligence training[1]. A part of bridging the gap between desired and available computing power are Ising machines[2]. Those specialized hardware implementations focus on solving optimization problems formulated as the Ising model, which are non-deterministic polynomial-time (NP)-hard and thus require intensive computation power to solve[3]. Researchers are exploring different physical implementations such as quantum computing[4], optics[5,6], memristors[7], and spintronics[8–10] to directly employ them for computation in the form of a highly efficient Ising machine. However, such systems are difficult to implement physically and are not yet ready for mass production compared to competing approaches using well-established complementary metal-oxide-semiconductor (CMOS) technologies. Besides specialized digital processors[11,12] the class of oscillator-based Ising machines (OIMs), which is usually implemented in CMOS as well, is very promising due to their high speed and energy efficiency. Systems capable of solving problems as fast as 50 ns at a low total power consumption of just 42 mW for a 1968 node system have already been reported[13]. Other systems with varying network topologies have been presented as well[13–19]. The efficiency and effectiveness of oscillator-based computing are also highlighted by the closely related oscillatory neural networks, which are used for machine learning tasks[20–22].

Well-known examples of NP-complete optimization problems are the traveling salesman, the knapsack problem, the maximum cut, and the remaining of Karp's 21 NP-complete problems[23]. Those can be converted within polynomial time to the Ising model[24]. A few exemplary applications for the Ising model are routing[25], image segmentation[26], and collaborative filtering[27]. There are many more practical applications in finance, economics, traffic management, computer-aided design, and machine scheduling in a QUBO form[28], which can be transformed into the Ising model. The Ising model, which was initially introduced to describe the spins of ferromagnetic material[29], is defined as shown in Eq. (1). The Hamiltonian H, which is a function of the discrete spins $\sigma_i \in \{+1, -1\}$ shall be minimized. $J_{ij} \in \mathbb{R}$, $i \neq j$ are the coefficients for the interaction between spins, while $h_i$ acts on just a single spin and gives a bias to the discrete state. In the following, we will refer to the term $\sum_{\langle i \rangle} h_i \cdot \sigma_i$ as 'bias'. The similar QUBO problem form is shown in Eq. (2), where $f_Q$ with $Q_{ij} \in \mathbb{R}$ shall be minimized by the discrete vector x with binary variables $x_i \in \{0, 1\}$. Both models can be transformed into each other as shown by conversion between $x_i$ and $\sigma_i$ in Eq. (3), where a $Q_{ij}x_ix_j$ QUBO term gets transformed into the Ising coefficients $J_{ij}$, $h_i$, and $h_j$. Consequently, we focus on the Ising model in the following since it is more commonly used in the field of OIMs.

$$H(\sigma) = -\sum_{\langle i,j \rangle} J_{ij} \cdot \sigma_i\sigma_j - \sum_{\langle i \rangle} h_i \cdot \sigma_i \tag{1}$$

Technical University of Darmstadt, Integrated Electronic Systems Lab, Darmstadt, Germany.
✉e-mail: Markus.Graber@ies.tu-darmstadt.de

$$f_Q(x) = \mathbf{x}^\top Q \mathbf{x} = \sum_{\langle ij \rangle} Q_{ij} \cdot x_i x_j \tag{2}$$

$$x_i = \frac{\sigma_i + 1}{2} \tag{3}$$

The concept of computation using OIMs is outlined in Fig. 1a. An optimization problem from an application must first be converted into the Ising form. This problem, consisting of discrete variables and weights linking those, is then embedded into the physical oscillator network. Each discrete variable must be mapped to a physical oscillator, which represents the binary states of a discrete variable in its phase angle, being either in-phase or anti-phase ($\pi$ phase shifted). Physical coupling circuits between the oscillators realize the weights of the optimization problem. After the successful embedding, the oscillators run and mutually influence their phases, which forms a solution to the optimization problem. Since phases are continuous while combinatorial optimization problems have only discrete states, a sub-harmonic injection locking (SHIL) is used before readout. The SHIL perturbs the oscillators and forces alignment of their phases in discrete groups. Details about the phase dynamics can be found in[30–32]. The phases are read out to assign the discrete states of the corresponding variables $\sigma_i$ based on their position in one of the two by $\pi$ separated distinct groups. An exemplary problem showing the letter 'G', which was computed on the actual chip, is shown in Fig. 1b.

Unfortunately, the graph embedding of the problem graph into a hardware connectivity network can be NP-hard as well[33,34]. Creating another NP-hard problem to solve the target NP-hard problem is one of the major drawbacks of such computing systems and can become a serious bottleneck of the overall computation. However, the embedding is a general issue of such hardware-based Ising machines with a sparse network, independent of the underlying physical devices. In general, it is sufficient to find just a single valid graph embedding into the available hardware network. Consequently, the hardware network should be designed to simplify the problem embedding to prevent it from becoming the bottleneck of the computation. While all-to-all network connectivity does not require time-consuming embedding, the number of connections $\frac{n}{2}(n-1)$ scales quadratically with the network size $n$, making it challenging for large networks. Besides the immense area, the coupling can get difficult. For example, the delay and

capacitive load introduced by the numerous connections poses a serious challenge. Embedding approaches often combine multiple physical nodes to form a single discrete variable to increase the connectivity[33,35]. However, this adds substantial overhead from the additionally needed physical nodes and their connections and can reduce the solution accuracy[15,35]. Hence, we propose a configurable, routable network with strong local connectivity as a trade-off. The idea is, that most of the graph edges should be embedded into the local connectivity. The routing network is just used for the remaining edges, which cannot be embedded locally. While the first edges can usually be easily mapped to the hardware connectivity graph, it gets increasingly difficult to find valid mappings for the remaining edges the more nodes and edges are already mapped to a hardware component. By using the flexible routing connections on demand for these remaining challenging connections, the difficulty can be reduced. Hence, the addition of such routable channels simplifies the embedding and can increase the success rate compared to a local network[36].

A very similar configurable topology is the field programmable Ising machine (FPIM) proposed by Jagielski et al.[37]. They map boolean satisfiability problems (SATs) to a hardware representation and use a modified open-source field-programmable gate array (FPGA) tool flow to embed such problems in the hardware. This flow enables them to precisely analyze the needed hardware connectivity to embed a given set of optimization problems.

## Results

The contribution of this work is as follows: We present a scalable architecture of an OIM system designed for solving Ising and QUBO problems with high accuracy and discuss the analog computing circuitry as well as the routing concept. We show extensive benchmarks solving maximum cut problems of the prototype, which is illustrated in Fig. 1c. Further experimental measurements highlight the importance of critical parameters like the coupling strength, SHIL strength, and system frequency for a high solution accuracy. Furthermore, the routable connections are evaluated and the calibration methods to compensate for unavoidable manufacturing mismatch as well as varying operating conditions like voltage, process, and temperature are discussed. Although abstract oscillator simulation models are available[38–41], they might not consider all non-idealities of real circuits. A more precise transistor-level simulation of the whole network is not feasible

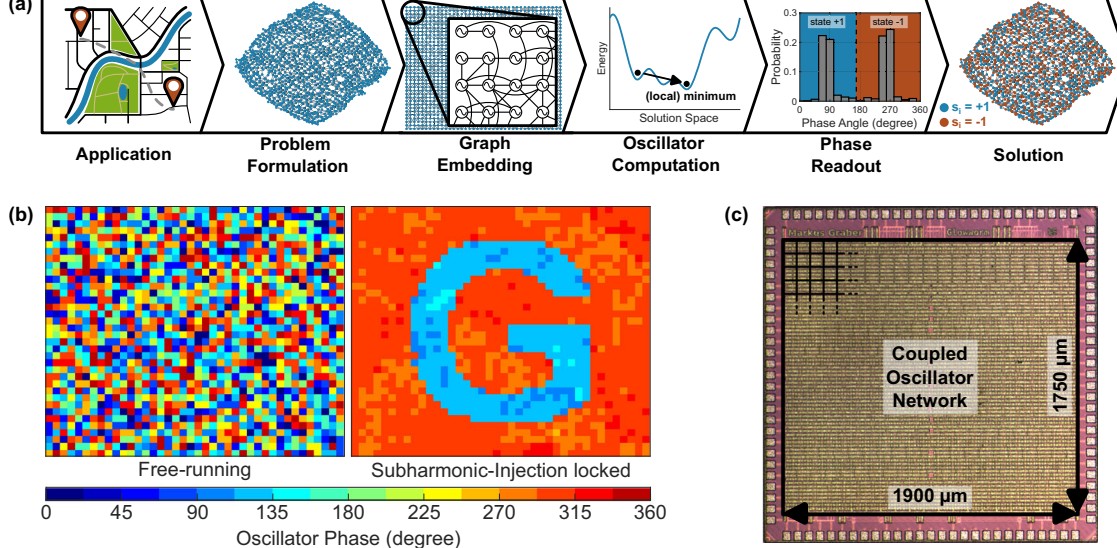

**Fig. 1 | Overview of computing with our OIM chip. a** Conceptual step-by-step procedure of computing with OIM systems. The application problem is formulated in the Ising form. After embedding this problem graph into the OIM-system network, the coupled oscillators naturally strive towards a (local) minimum state in hardware. The solution is finally obtained by reading out the phases. **b** Exemplary outcome of the computing process in the phase domain for a created illustration problem showing the letter 'G' of the chip codename 'Glowworm'. **c** Die micrograph of the 1440 coupled oscillator network implemented on a core area of 3.325 mm². The power supply distribution using the topmost layers hides the circuitry below.

because of the excessive simulation runtime. Hence, this work is an experimental approach to investigate and verify the operation of OIMs.

## System

The connectivity of the system network is shown in Fig. 2a. Each oscillator has a local connectivity to 11 neighbors forming a fixed, sparse network. It connects to the horizontal, vertical, and diagonal neighbors in a 2-dimensional grid, but also to the neighbor of the horizontal and vertical neighbor. The configurable routing network (blue) with the Wilton-style switch blocks[42] provides 4 connections per oscillator to any other. To implement the bias term $h_i$ of the Ising model the oscillators can be coupled to a globally distributed phase reference (red). Our 4.6 mm² prototype manufactured in a 28 nm technology of a world-leading foundry contains a total of 1440 oscillators and 11,724 coupling elements. 2584 of these connections are freely routable across the whole chip, while the remaining have predefined local connectivity.

**Node Implementation**. The block diagram of an oscillator node representing a discrete variable $\sigma_i$ is shown in Fig. 2b. The central element is the ring oscillator shown in Fig. 2c operating at a frequency of 100 MHz, which keeps signal propagation delays to neighboring oscillators still small compared to the oscillation period. Due to the necessary graph embedding and data transmission for the OIM-system configuration, we think that higher oscillation frequencies will have limited benefit for the overall computation. We use a differential delay cell design shown in Fig. 2d, which was empirically optimized for its high sensitivity to coupling currents. The small number of 4 delay cells provides a good

sensitivity but still sustains oscillation at strong injected currents reliably. The differential architecture provides phase shifts of $\pi$ by swapping both differential signals. This is used to implement in-phase (0-phase difference) and anti-phase ($\pi$-phase difference) coupling. Additionally, the higher power supply rejection of differential designs reduces unwanted coupling via the power supply. However, a differential to single-ended converter provides the oscillator signal for the all-digital phase-to-digital converter (PDC)[43]. Although just a binary phase measurement would be sufficient for the Ising model, a 4-bit resolution is implemented to investigate the OIM computation in more detail. The sensitive oscillator waveform is buffered to avoid loading effects[44]. The configurable coupler circuits convert the phase differences between the oscillators into a current, which is summed up and injected into the oscillator to change its phase. An all-transistor coupler design is proposed in this work to avoid large resistors or capacitors. As shown in Fig. 2e it consists of two differential pairs to modulate the injected current based on the phase difference. Swapping the input signals implements the sign of the weights by coupling in-phase or anti-phase. By setting the bias current via an individual 4-bit digital-to-analog converter (DAC) for each coupler, the weight coefficients of the optimization problem are set. The current summation circuit as shown in Fig. 2f accumulates the current of all couplers for injection into the oscillator, while its self-biasing stabilizes the common mode voltage level. In combination with the output buffer of the oscillator, which mitigates the capacitive loading caused by the connected couplers, several coupling connections can be simply added to achieve the desired connectivity. To implement the bias term $h_i$, we couple the oscillators to the already available reference clock, which adds

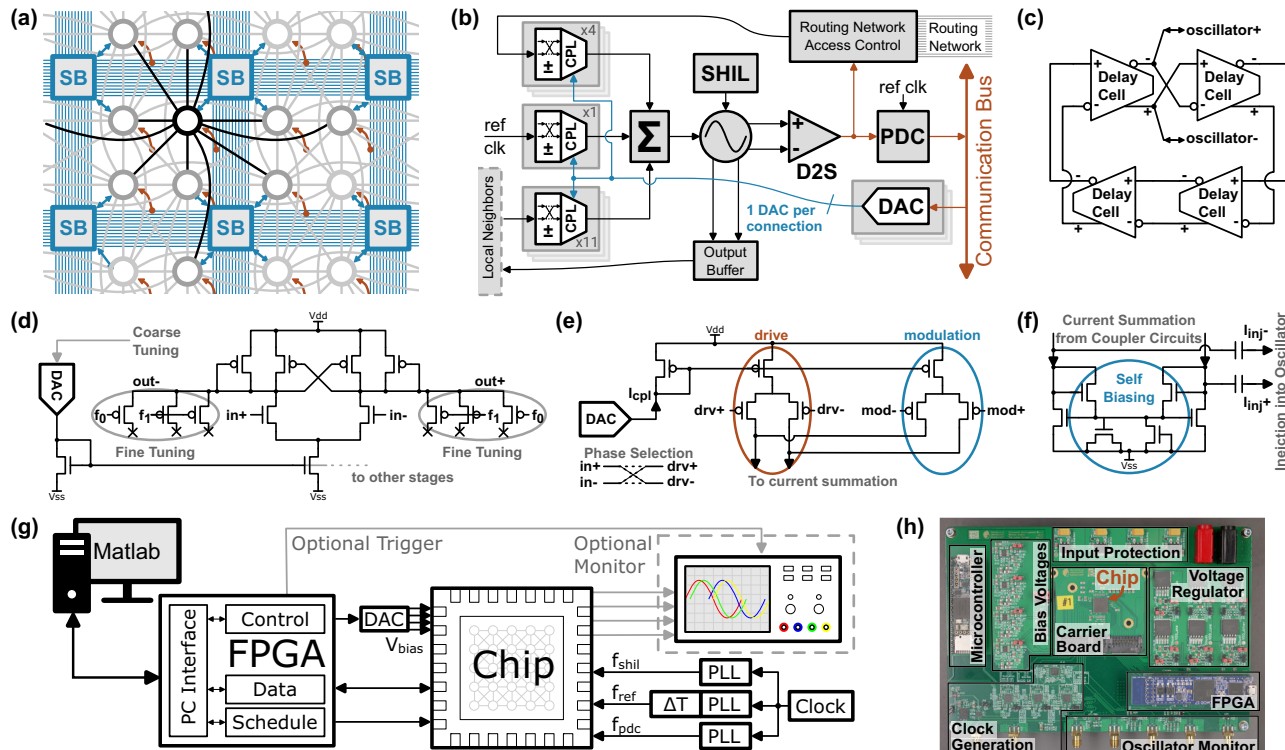

**Fig. 2 | System structure, circuits, and test setup. a** Simplified excerpt of the oscillator network. The local connectivity of a single node is highlighted in black, the routing network with the switch blocks (SB) is blue, and the reference connections for the external bias term are red. The local connectivity of other nodes is shown in gray for better visibility. **b** Simplified block diagram of an internal node, which represents one discrete variable of the optimization problem. CPL: Coupler Circuit; D2S: Differential-to-single ended converter; DAC: Digital-to-analog converter; PDC: Phase-to-digital converter; ref clk: reference clock; SHIL: Sub-harmonic injection locking **c** 4-stage differential ring oscillator. **d** The delay cell of the oscillator

including a coarse and fine-tuning for the frequency calibration. **e** Simplified schematic of an unidirectional coupler stage, which is used back-to-back to establish the bidirectional coupling connection. Each differential pair has a drive (drv) and modulation (mod) input. **f** Current summation circuit, which combines the currents of each coupler and injects the sum $I_{inj}$ into the oscillator. **g** Overall computing and test concept, which is managed by a custom Matlab application, consisting of an field-programmable gate array (FPGA), external clock generation using phase-locked loops (PLLs), delay line ($\Delta T$), and optional oscilloscope. **h** Photograph of the test board with the pluggable chip carrier board in the center.

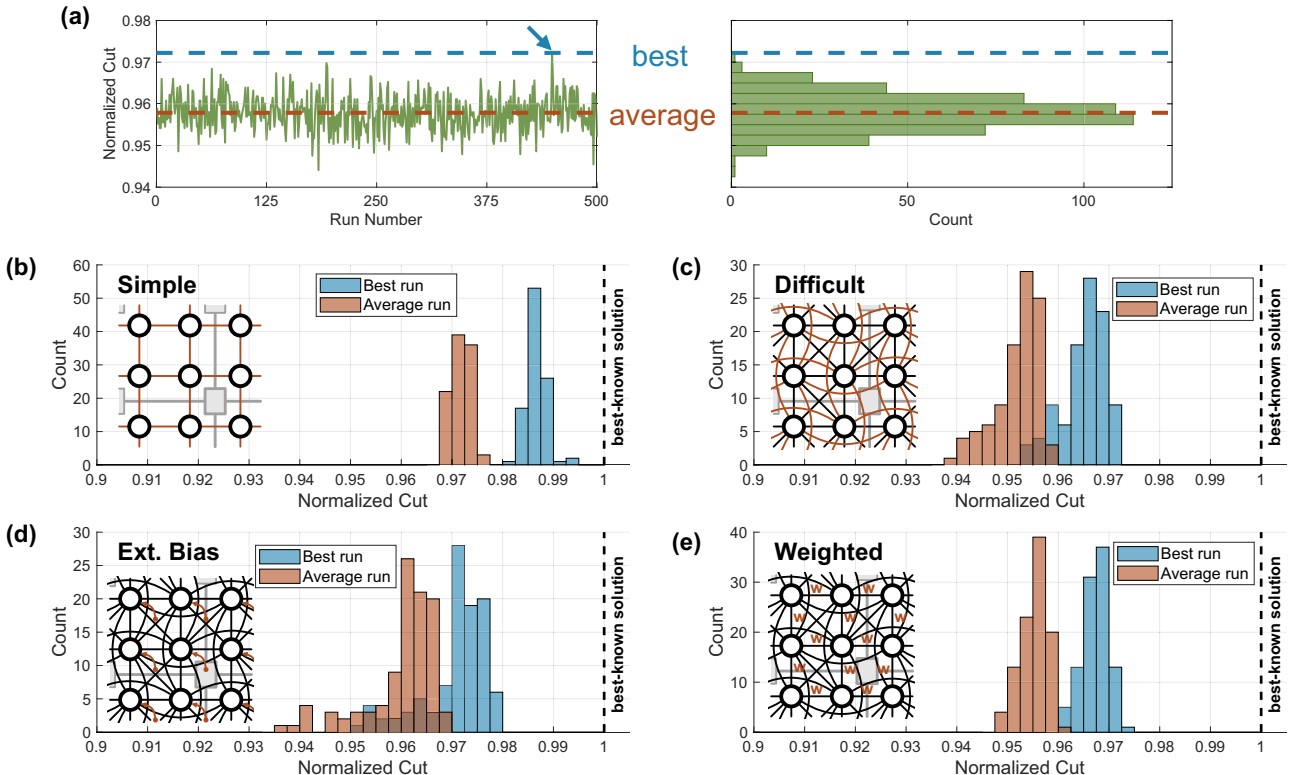

**Fig. 3 | Benchmarks with different optimization problem sets of our OIM (for more details see "Benchmarks" in methods).** Each problem set contains 100 problems, which are repeatedly solved 500 times each. **a** Example of solving the same problem repeatedly 500 times to record the best and average cut. **b** 'Simple'[47]

benchmark set with horizontal and vertical neighbors. **c** 'Difficult'[48] benchmark set using all 11 available local neighbors. **d** 'Bias'[49] benchmark set which additionally includes the external bias term $h_i$ respectively $Q_{ii}$. **e** 'Weighted'[50] benchmark set, which additionally uses weights with random 4-bit resolution for all edges.

just a small area overhead. The SHIL-injector gradually applies the perturbating SHIL signal to gently freeze the oscillator phase and therefore discretize the oscillator phases in the two spin groups. This ensures, that the coupled oscillators try to minimize the Ising Hamiltonian[30]. Based on the experimental evaluation of our previous generation with adjustable ramps, we decided to ramp up the SHIL in 8 linear steps, which are increased after each nominal oscillation period.

**Routing.** The routing enables the connection of two arbitrary oscillators on the chip together if suitable routing tracks are not already occupied. When transmitting a signal through the routing network, it gets inevitably delayed by a few hundred picoseconds up to a few nanoseconds depending on the distance. This delay becomes particularly relevant at the oscillation period of 10 ns. The received time-delayed signal creates a perceived phase lag at the distant oscillator, that does not match the actual phase of the receiver. When coupled, the oscillators will react based on this perceived phase potentially causing a runaway situation and a frequency shift. To address this, we have introduced a delay compensation scheme, the design and simulation of which are discussed in our previous work[45]. This method essentially transmits the oscillator signal of the previous stage, which can be seen as the oscillator waveform with a "negative" delay. An adjustable delayline supplements the propagation delay through the routing network to match that "negative" delay so that the received oscillator waveform is time synchronous with the actual transmitting oscillator. So, the perceived phase at the receiving oscillator matches the phase of the transmitter. However, the phase dynamics of the coupling are altered, because any response is perceived at the receiver with a full period delay. Because the propagation delay heavily depends on the traveled distance but also on temperature, process, and supply voltage, the optimal delayline setting is obtained using a calibration, which is discussed later in the calibration section.

**Computing & benchmark setup.** The overall system concept of our computing setup is shown in Fig. 2g and its physical realization is depicted in Fig. 2h. The OIM chip just requires external clocks and an FPGA, which could be in general integrated on the same die as well. The data transfer and generation of the timing signals are handled by that FPGA, which interfaces with a computer. Matlab scripts handle the test management, OIM configuration, timing scheduling, and analysis of the results. Four off-chip drivers allow us to investigate and verify the behavior of selected oscillators using an external oscilloscope, which is not needed for normal computing operation. External bias voltages are used to adjust the oscillator frequency, SHIL-injection strength, and global coupling strength. A single computation takes in total 950 ns. The first 200 ns are used to let the oscillators couple. Then the SHIL strength is ramped up to gradually force the coupled oscillators into the two phase groups, which takes 182 ns including a small margin for stabilization. Lastly, the phase measurement to obtain the computed result takes 568 ns. The energy consumption per computed problem varies between 404 nJ and 466 nJ across the individual problems of the later discussed 'difficult' benchmark set. A single oscillator including its periphery has a measured power consumption of 113.3 μW and a coupling connection needs approximately 23 μW.

## Benchmarks

To evaluate the accuracy, we solve randomly generated benchmark problems and compare them against the best-known solution. Multiple network topologies of varying difficulty are tested, which are sketched together with the histograms in Fig. 3. The outcome of an OIM computation using our chip depends on the random initial states and might be affected by noise, so that the computed solution varies randomly when computing the same problem again. As shown in Fig. 3a, we solved each problem 500 times and noted the average (red) and best (blue) solution. The computed cut is

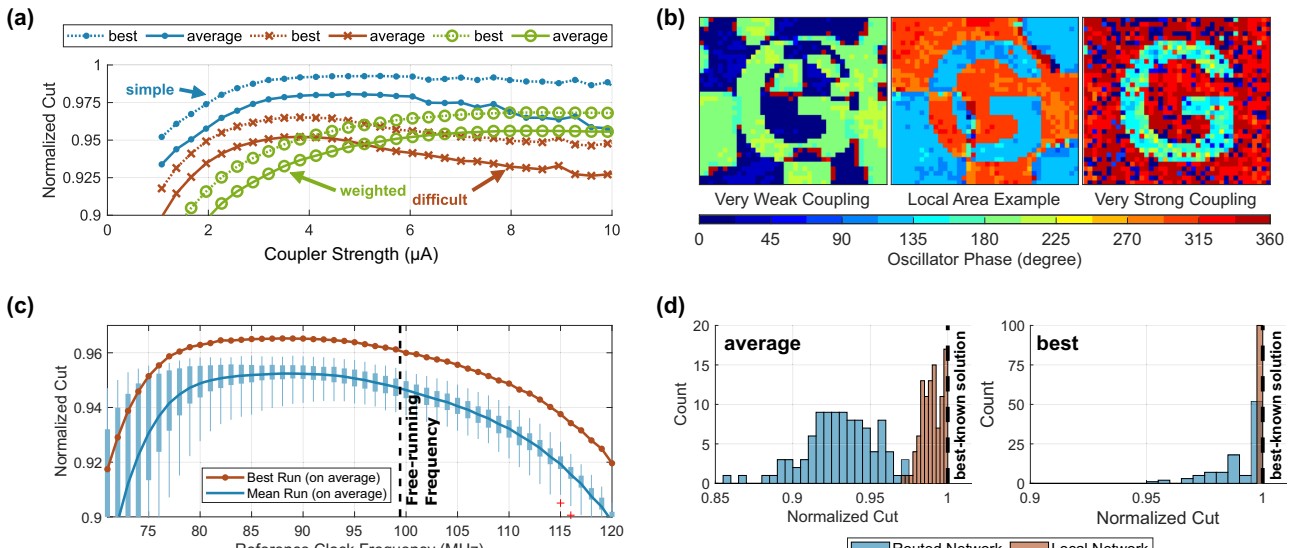

**Fig. 4 | Experimentally measured analysis of the coupling parameters. a** The measured impact of the coupling strength on the solution accuracy. **b** Visual example of the impact of weak and strong coupling as well as remaining formation of local areas at the optimal coupling strength. **c** The experimental measured impact of the reference clock frequency on the solution accuracy. The boxplots indicate the minimum, 25% percentile, 75% percentile, and maximum of the individual problems of the 'difficult' set. **d** Comparison of the solution accuracy between an identical problem solved with a purely local connection and a purely with the routing network. The average accuracy is shown on the left and the best run for each problem is on the right.

normalized with the best-known solution, which was obtained using the commercially available software Gurobi[46]. Each of the 500 runs per problem exhibits a unique solution, where at least 6.4% of states are different compared to every other solution of the same problem. When running the oscillators without any mutual interaction, only 0.18% of oscillators show a weak correlation between their neighbors. Furthermore, no auto-correlation between the previous phases of an oscillator can be detected after roughly 1200 ns.

The 'simple'[47] set shown in Fig. 3b contains only connections to vertical and horizontal neighbors on a two-dimensional grid (0.27% of all-to-all connectivity). Since all weights of a maximum cut problem have a positive weight of 1, a checkerboard pattern is a maximum solution, with the cut being the sum of all edge weights. While such a problem might be simple for an iterative algorithm, the parallel nature of OIM systems does not benefit much from this structure. However, the optimum solution of a checkerboard pattern satisfies all coupling connections, which makes this problem less difficult for OIMs than the others. The average cut reaches more than 96.8% of the best-known solution. The 'difficult'[48] set in Fig. 3c uses all available local connections (0.73% of all-to-all connectivity) while keeping the weights at positive 1. The denser local connectivity makes it more difficult to solve since there is no contention-free solution anymore, which satisfies all coupling connections. To emphasize the higher difficulty, computing the reference solution using the commercial software Gurobi[46] for the 'difficult' set took on average 10,000x longer than the 'simple' set. However, this is just an indication and no proper measurement of difficulty. As expected, the solution accuracy is worse with an average normalized cut of better than 94%. The 'bias'[49] set in Fig. 3d adds the bias terms $h_i$ to the structure of the 'difficult' problem set, which shows no accuracy degradation from the added bias term. While the previous problems have a fixed weight of +1, the 'weighted'[50] set tests the same topology with the full 4-bit weight resolution in Fig. 3e. The performance is very similar to the 'bias' benchmark set. Interestingly the spread between individual problems is reduced. The parameters like the coupling strength were individually tuned for all four shown benchmark sets to achieve the best accuracy.

### Experiments & optimization

Figure 4a illustrates the impact of the coupler strength, which can be globally scaled using an external bias voltage, on the computing performances by evaluating the 'simple', 'difficult', and 'weighted' benchmark sets at different coupling strengths. A distinct range for the coupling strength exists, which achieves the best performance but depends on the benchmark problem set respectively problem connectivity. Figure 4b graphically illustrates the impact of the coupling strength using the already known example from Fig. 1b. A weak coupling strength causes multiple local areas, which are solved in themselves correctly, but exhibit a distinct boundary to other such local areas. This effect is much less pronounced at higher coupling strengths, but still relevant around the best coupling strength. At higher coupling strengths, the forming of such local areas is less likely, but faulty spots appear, which couple in an undesired phase despite the clear coupling inputs from the neighbors. So, the coupling strength and SHIL can be thought of as the equivalent of an annealing schedule, which is frequently used in simulated annealing[51]. We suppose, that similar mechanisms considerably reduce the accuracy of the OIM for most problems. It should be emphasized, that the choice of coupling strength is very system and problem-specific. Therefore, we recommend that those parameters are not fixed by design, but remain adjustable during operation. Figure 4c shows a sweep over the reference clock frequency, where the SHIL frequency is exactly double that value. The best results are achieved for a frequency between 80 and 90 MHz, which is 10% to 20% lower than the actual oscillator free-running frequency. To assess the effectiveness of routable connections, which have slightly different phase dynamics due to their inevitable routing delay with compensation scheme[45], a comparison is provided in Fig. 4d. A reduced 10 × 10 network size 'simple' benchmark set was solved with routable connections and for comparison using local connections. While the local couplings find the best-known solution for all problems, the routing only for 52 out of the 100 problems. The spread between the problems widens as well, where the worst problem achieves just 85% using the routing network compared to 95% for the local routing.

### Calibration routines

The OIM system is a mixed signal design, where the analog components are responsible for the actual computation. The digital circuitry configures and supports the analog computing core. Unfortunately, random manufacturing mismatches and imperfections influence the sensitive analog circuitry. For example, some transistors might have a higher conductivity than others, causing differences in the frequencies. We employ an on-chip frequency

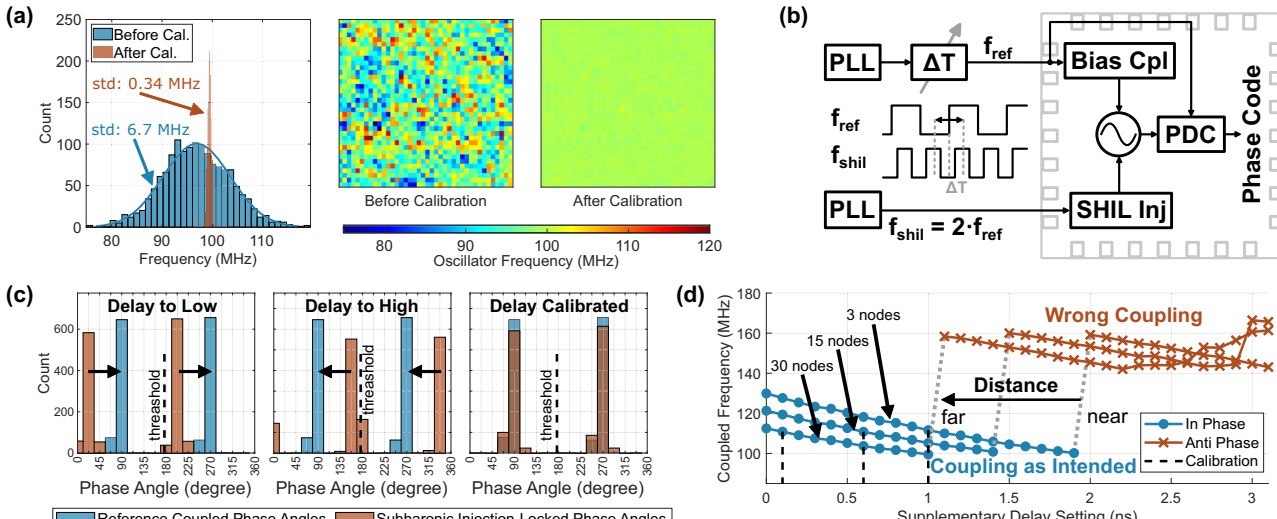

**Fig. 5 | Calibration principles used to counteract manufacturing imperfections.** **a** Distribution of the oscillator free-running frequencies across the chip before and after the calibration. The color-coded maps show the spatial distribution across the whole chip. **b** Principle of the external delay adjustment for the phase calibration. Bias Cpl: Coupler for spin bias; PDC: phase-to-digital converter; PLL: Phase-locked loop; SHIL Inj: Sub-harmonic injector; $\Delta T$: delay-line; **c** Alignment procedure of the reference and SHIL-clock by adjusting the external clock delayline $\Delta T$. **d** Frequency to delay relationship of the routing connections, which is the basis of the routing calibration.

calibration, a Matlab-assisted phase calibration, and a Matlab-assisted routing delay calibration to compensate for such manufacturing mismatch.

**Frequency calibration.** The frequency of the oscillators is approximately Gaussian distributed with a spread between 75.6 MHz and 119.1 MHz within a silicon die as shown in Fig. 5a. Since such free-running frequency differences cause phase offsets in the coupled state, which can affect the computation[44], a fine and coarse calibration adjusts the individual frequencies to the target 100 MHz. After calibration, the standard deviation of the oscillator frequency is reduced from 6.7 MHz to just 0.33 MHz, which improves the computing accuracy on average by approximately 0.7 percentage points. According to our previous chip generation, any frequency deviation less than approximately 1 MHz yields negligible performance gains.

**Reference phase calibration.** While the weights between two discrete variables ($J_{ij} \cdot \sigma_i \sigma_j$) just depend on the difference between two discrete states, the bias terms ($h_i \cdot \sigma_i$) depend on just one state, which requires an absolute phase reference to define the actual states. The calibration principle is shown in Fig. 5b, which adjusts the external delayline $\Delta T$ to change the phase between the reference clock and the SHIL clock. The distribution of the reference coupled oscillators, with one half coupled in-phase and the other half anti-phase, is shown in Fig. 5c (blue). The distribution is only slightly influenced by the operating frequency and coupling strength, as well as supply voltage, temperature, and process. The phase in the SHIL locked states (red) is then adjusted using the delayline $\Delta T$ to match the reference coupled phase distribution (blue). Without calibration, the oscillators might not couple as intended by the bias term ($h_i \cdot \sigma_i$) leading to poor accuracy. The phase threshold can be determined by finding the center of both reference phase (blue) groups. The phase angle of the +1 spin group can be clearly identified by the phase of the in-phase coupled oscillators.

**Routing delay calibration.** Any signals, that are transmitted through the routing network, experience a propagation delay. This is greatly affected by the spatial distance between the connected oscillators as well as supply voltage, temperature, and process variations. Unfortunately, the coupled oscillators are sensitive to that delay leading to changes in their frequency and phase. They either couple as intended or settle in the wrong, opposite

phase (shifted by $\pi$). This observed behavior is a probability function depending on the delay instead of a sharp transition. The measurements in Fig. 5d show the impact of the routing network on the frequency and phase for a pair of coupled oscillators at three different distances. The delay is changed using a delayline with a 100 ps step size, which is implemented for the calibration scheme. The plot is simplified for better clarity by showing the behavior of the majority of oscillators instead of the actual probability density, which is the reason for the distinct transition. While the frequency shows a declining trend with rising delay, the oscillators couple in the wrong phases and increase their frequency by approximately 50% when exceeding a certain threshold. For calibration, we choose a delay setting leading to a coupled frequency closest to the free-running frequency, which still couples as intended with a probability greater than 70%. This delay value, which is separately obtained for all possible routing distances, is stored in a lookup table. Without the calibration (e.g., setting the delaylines to 0 for their minimum delay), the routing channels essentially fail to function, achieving a similar performance as a random number generator. However, the delay setting has a roughly 400 ps tolerance range, which exhibits very similar performance.

## Comparison with Other Works

A comparison with other integrated CMOS OIMs is provided in Table 1. This work offers a versatile trade-off and introduces routable connections between oscillators. It has a competitive number of nodes and edges. Together with Lo et al.[15], it is the only system to provide the bias term of the Ising model. However, our approach can be used for any network topology instead of requiring an all-to-all connected oscillator. While all-to-all connectivity is superior, they are limited to very small systems with just 30[16] and 48[15] nodes. The routing channels in this work add great flexibility to the network. In combination with the local connectivity, it is well-suited for large, sparse problems. In comparison to our predecessor[19], we managed to maintain the high solution accuracy despite increasing the number of nodes and edges while reducing the power and area per node.

## Discussion

We presented a scalable approach for OIMs capable of solving optimization problems in Ising as well as QUBO formulation. The critical characteristic of such OIM systems is the network connectivity. For solving real-world optimization problems, the network should be as large as possible having

**Table 1 | Comparison of integrated CMOS OIMs including the predecessor as "Graber et al.[19]"**

| | Bashar et al.[16] | Ahmed et al.[17] | Mallick et al.[18] | Moy et al.[13] | Lo et al.[15] | Graber et al.[19] | This work |
|---|---|---|---|---|---|---|---|
| Technology | 65 nm | 65 nm | 65 nm | 65 nm | 65 nm | 28 nm | 28 nm |
| Number of nodes | 30 | 560 | 600 | 1968 | 48 | 400 | 1440 |
| Number of edges | 435 | 1,585 | 29,076 | 7,607 | 1,236 | 1,482 | 11,724 |
| Weight resolution | fixed | fixed | fixed | 5 levels | ≈ 4-bit + sign | 6-bit + sign | 4-bit + sign |
| Topology | all-to-all | hexagonal | intra/inter tile | king's graph | all-to-all | king's graph | local + routing |
| Edges per node | 29 | 6 | 111 | 8 | 47 | 8 | 15 + 1 |
| Coupling principle | capacitor | inverter | capacitor | transmission gate | transmission gate | DAC driven stage | DAC driven stage |
| Ising model support | $J_{ij}$ | $J_{ij}$ | $J_{ij}$ | $J_{ij}$ | $J_{ij}$ and $h_i$ | $J_{ij}$ | $J_{ij}$ and $h_i$ |
| Operating frequency | 45 kHz | 118 MHz | 45 kHz | 1 GHz | ≈ 28.5 MHz | 50–200 MHz | 100 MHz |
| Computing time | - | 200 ns[1] | 6.4 ms | 50 ns | ≈ 2–14 µs | 713 ns | 950 ns |
| Solution accuracy[5] | - | ≥82% | ≈ 80%[3] | up to 95%[2] | NR[4] | ≥95% | ≥94% |
| Power | 1.76 mW | 23 mW | 25 mW | 42 mW | 16–105 mW | 302.9 mW | 460.3 mW |
| Chip area | 1.44 mm² | 1.44 mm² | 4 mm² | 2.1 mm² | 1.8 mm² | 2.2 mm² | 4.6 mm² |

[1] simulated; oscillator coupling only.
[2] with additional digital post-processing.
[3] oscillator network performance only, digital post-processing excluded.
[4] reported accuracy not compatible.
[5] Accuracy measurement methodology varies between publications.

tens of thousands of nodes with as many connections as possible. However, the quadratic increase of connections in all-to-all networks makes a circuit implementation unfeasible for large systems due to the immense area demand. Our approach unites these two opposing requirements by introducing flexible, routable connections between oscillators. This is a good trade-off for sparse optimization problems as it provides every possible connection between nodes while avoiding hardwiring every connection to save area. Additionally, the flexibility of the routing simplifies the computationally intensive embedding of the optimization problem into the OIM network, which can potentially exceed the computation time of the actual problem.

Our fabricated chip demonstrates the effectiveness of OIMs by solving max-cut problems within 950 ns while consuming an average power of 460.3 mW. The computed solutions reach on average at least 94% of the optimum and even more when selecting the best runs. The two major design challenges are area efficiency and random device variation from manufacturing. To be as compact as possible, we proposed a fully transistor-based coupler circuit with weights set by simple current DACs to avoid large, area-intensive resistors and capacitors. A time-multiplexing of the DACs, which occupy 12.8% of the core area, could further save area. To deal with process variation and mismatch of individual devices, we use calibration schemes and fine-tune parameters such as coupling strength and SHIL frequency. Our experiments emphasize that good performance is reached in just a narrow range of the coupling strength, which, however, shifts with the optimization problem topology. The oscillator frequency calibration increases the accuracy by approximately 0.7 percentage points. A software-based phase angle calibration ensures correct operation of the bias term '$h_i$' of the Ising model, which is area-efficiently implemented utilizing the already distributed reference clock for the phase measurement. The calibration of the routing delay is crucial. Without it, the network is not usable and just as effective as a random number generator. Unfortunately, despite the compensation scheme including calibration, the routing connections exhibit lower performance due to the unavoidable propagation delay than native local connections between neighboring oscillators.

A calibration scheme for the coupling weights might further increase the accuracy of OIMs. Additionally, a digital post-processing of the measured phases could improve the performance as well. While our approach is well suited for large, sparse optimization problems, we think that large and dense problems might likely not be feasible for OIMs due to the high amount of needed oscillators and coupling links. Instead, hybrid approaches, which

could partition a large graph into smaller sections to be solved by OIMs, might combine the advantages of OIMs with digital processors.

## Methods
### Chip design
The chip was designed in a 28 nm node of a world-leading semiconductor foundry. A 4.6 mm² silicon chip was manufactured using the mini@sic program of the Europractice IC service. The latest process design kit from the foundry was utilized, providing compact models for simulation. The design uses regular core devices operating at 0.9 V with different thresholds. Cadence Virtuoso version IC6.1.8 was used for schematic and layout entry. Cadence Spectre 21.1.0 was used for transistor-level circuit simulations. Siemens Calibre nmDRC, nmLVS, and PEX version v2020.4_15.9 are used for layout rule checking and parasitic extraction. The digital control logic was synthesized utilizing the foundry-provided standard cells with Synopsys Design Version R-2020.09-SP2 and Cadence Innovus 21.10 for the place-and-route. More details about the overall system implementation can be found in our previous work[45,52].

### Delay-line calibration
The delayline calibration is explained in detail in our separate paper[45]. For convenience, the principle is outlined in Fig. 6. As shown in Fig. 6a the internal waveforms of the oscillator are phase-shifted. So the output at '4' is a delayed version of the previous stage at '3'. The output of this previous stage is transmitted through the routing network as shown in Fig. 6b. The unavoidable delay $T_{routing}$ is supplemented with the adjustable delayline $\Delta T$. So, the received phase at oscillator B (point '3') can be aligned with the actual output of oscillator A (point '4') to minimize the skew $T_{skew}$. Since $T_{skew}$ cannot directly be measured and varies with the operating conditions, the discussed calibration is performed.

### Chip reconfigurability & external adjustments
Every local coupling connection on the chip can be configured independently. Every weight is set using a 4-bit DAC value and an additional bit to select between in-phase and anti-phase coupling. The same applies to the routing connections, which have additionally a 5-bit word for the delayline with approximately 100 ps steps. The coupler for the bias term $h_i$ is similar, but has 5-bit resolution and has four times the maximum strength of the regular $J_{ij}$ connections. Every switch of the switch block and the switches to connect the oscillator with the routing network can be individually enabled/

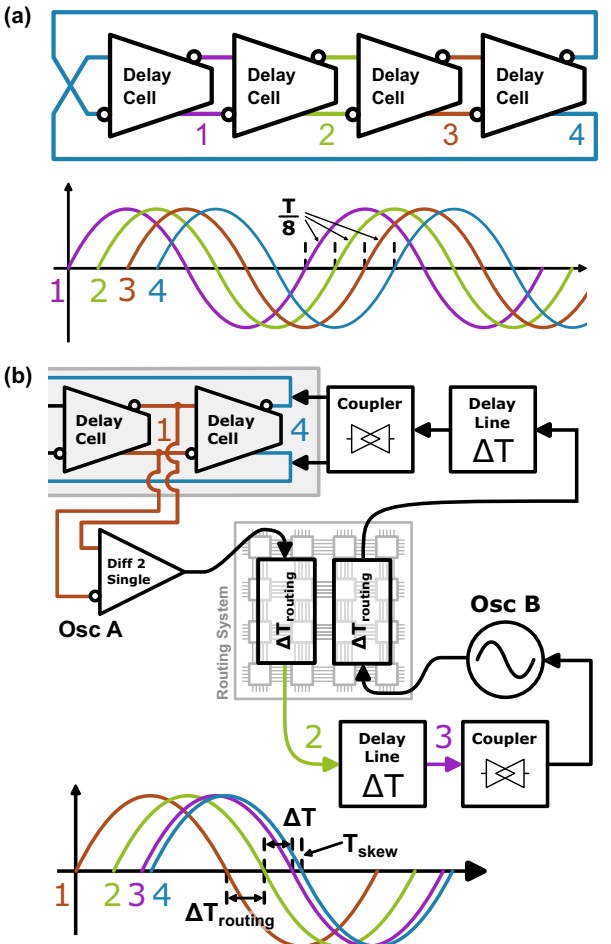

**Fig. 6 | Principle of the used delay compensation scheme taken from our previous paper**[45]**. a** Phase shift internal of the oscillator. **b** Compensation scheme showing the transmitting of a signal from oscillator (Osc) A to B. The unavoidable routing delay $T_{routing}$ is supplemented by the adjustable delayline $\Delta T$ so that a small skew $T_{skew}$ is reached at the receiving oscillator B.

disabled. The algorithm to configure these switches is mentioned below in section 'Path Routing'.

External analog bias voltages are internally converted into bias currents using resistors. So, the coupling strength, the SHIL-strength, and the oscillator frequency can be externally set. Furthermore, the SHIL frequency can be set with the external phase-locked loop (PLL). The timing to control the operating states (enable oscillator couple, apply the SHIL signal, phase measurement, oscillator measurement, frequency calibration) is triggered by the external FPGA (Fig. 2g) on the test printed circuit board (PCB) (Fig. 2h).

## Experimental results

The dies with 108 pads, whereof 67 pads are for power supply and bias voltages, were wire-bonded into a QFN-64 package and soldered on a pluggable carrier board. The presented results are from die number 1, the first one we have soldered on a carrier board without any selection. The benchmarks were conducted with the other 9 chips as well. The test PCB contains all circuitry for the experiments, just an external power supply was used additionally. A Matlab script coordinates the testing by selecting optimization problems, generating the configuration bitfiles for loading the problem into the OIM chip, and configuring the test PCB. The interface between the computer and FPGA limits the maximum data throughput to 12 Mbit in our experiments. Therefore, the OIM system is disabled during

the configuration data transfer from the computer to the chip. During transmission of the measured phase data, the oscillators are kept frozen in SHIL-lock and stay in the same phase for sufficient SHIL strengths, which was validated regularly. We have no control over the initial phases at the start of the computation. All experiments were conducted at room temperature between 20 and 25 °C. The presented calibration data is averaged over 500 samples per oscillator. The routing frequency-delay behavior in Fig. 5d is the averaged behavior of 34 oscillator pairs sampled 500 times each.

**G-Letter example problem.** The example of the 'G' is generated from a black-and-white image of the letter G with identical resolution as the OIM network ($40 \times 36$), so that each pixel corresponds to one oscillator. It is iterated through all pixels in the image and the coupling connections are set based on the color of the direct horizontal, vertical, and diagonal pixel neighbors. If the color of the neighbor pixel is identical, a connection between the corresponding oscillators to couple in-phase is added. If the color of the corresponding neighbor pixel is the opposite, then an anti-phase coupling connection is inserted. Overall, the underlying problem is of a king's graph structure with all connections enabled, where the sign is determined by the color of the root image. By construction, all oscillators can couple in phases as intended, which is the optimum solution.

## Phase Threshold

The threshold to distinguish between the spins $\sigma_i = +1$ and $\sigma_i = -1$ state for all shown results was determined by testing all 16 possible thresholds (resolution of the phase angle by the 4-bit PDC) and selecting the threshold with the best Ising Hamiltonian. We have chosen this brute-force method because it ensures a repeatable and reliable calculation of the Ising Hamiltonian independent of any prior calibration or results. The test PCB generates the SHIL clock $f_{shil}$ and reference clock $f_{ref}$ from the same clock signal (see Fig. 2g) using Analog Devices ADF4351 PLLs. Since the internal oscillator feedback loop within the PLL is used for better jitter performance, we do not have control of the phase between $f_{shil}$ and $f_{ref}$. Any time the PLL setting is changed, that phase difference and hence the phase threshold changes. For non-experimental usage, one could either determine the threshold just once based on the phase values or use the reference calibration.

## Calibrations

The on-chip frequency calibration is conducted at every start of our test system because the previous calibration is lost after a power cycle. The calibration mainly compensates for random manufacturing mismatch and stays similar between calibration cycles. The phase calibration is conducted once at the start-up of the system and when the operating parameters are notably changed. This is the case, if either the PLL configurations are altered or the SHIL strength/coupling strength is considerably changed. The delay settings of the routable connections were calibrated once and the necessary delays depending on the length stored in a lookup table. This table was then used to get the delay settings in all following experiments.

## Benchmarks

The benchmark problems were randomly generated by a custom Matlab script tailored for our OIM chip network. To ensure that the generated problems follow the intended network topology, a list with all allowed and physically available edges is generated first. Then edges are randomly deleted until the desired network density is reached, which is randomly selected between 30% and 70% of all edges of that chosen topology ('simple': 841 to 1963 edges, 'difficult': 2285 to 5331 edges, 'bias': 2717 to 6339 edges, 'weighted': as difficult). No checks for the existence of isolated sub-graphs were applied. The edge weights (if applicable) were randomly chosen from a uniform distribution. We interpreted the problem as a maximum cut similar to other works, which closely follows the Ising Model. We extended the

maximum cut to include a bias term, which can be interpreted as an edge from a node to a fixed set. The edge weights of the maximum cut problem are limited to positive only values (transformed into negative $J_{ij}$ coefficients of the Ising model) for better comparability of the obtained cut. Consequently, the obtained cut is limited to the range between 0 and the best-known solution, which we normalize to 1. The sign of the weights in the maximum cut is simply inverted for transformation into the Ising model, so the OIM operates with just anti-phase coupling. Due to the differential design, where the in-phase and anti-phase coupling is achieved by just swapping the inputs to the coupler circuits in its multiplexer, the general coupling behavior stays identical. The best-known solution for each generated problem was obtained using the commercial tool Gurobi[46]. Gurobi version 10.0.3 was executed on an AMD Threadripper Pro 5965WX 24-core processor and 256 GB RAM without any time restriction, which took up to 4.67 hours per problem. The solution was guaranteed to be the optimal solution.

## Path routing

The connections through the routing network were configured using Matlab. It is based on a graph representation of the available hardware network, which includes every physical track and switch of the switch block. The 'shortestpath' function in Matlab, which internally uses the Dijkstra algorithm, calculates a path to connect the two oscillators. The returned path includes all physical tracks and switches of the switch blocks. These used elements are then marked as occupied so that they are avoided for the following paths. The obtained path is checked to be physically possible and verified to not contain any already occupied tracks. All routing connections of an optimization problem are applied in a sequential order, where the path for each connection only considers the occupied tracks from previous connections. For the routing problems in Fig. 4d, this approach connects the oscillators in the shortest, direct way.

## Power measurement

The test PCB (Fig. 2h) contains onboard current shunts for every supply and bias voltage rail, which are placed within the feedback loop of the onboard voltage regulator to compensate for the voltage drop over those shunts. A Keysight DAQ970A sequentially scans the voltage drop over the shunt resistors of the bias rails, analog supply, and digital supply. The separate supply rails for the I/O-ring and off-chip driver are excluded from the measurement. The operating state of the coupled as well as SHIL-locked oscillators were extended for multiple seconds to precisely measure the (average) power consumption. The power consumption during the SHIL ramping and phase measurement could not be directly measured due to their short duration but was approximated by the power consumption under SHIL lock. Additionally, the time and energy for writing the configuration data in the chip and sending the phase data out of the chip are omitted. The power consumption varies substantially with the optimization problem since they influence the amount of enabled oscillators and couplers. Thus, we recorded the power consumption for each problem of the 'difficult' set. Additionally, the total power consumption depends on the frequency and the global coupling strength, where the optimal parameters for the optimization problem were used.

## Data availability

The used benchmark problems 'simple'[47], 'difficult'[48], 'weighted'[49], and 'bias'[50] are available at figshare.

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

## Acknowledgements

We want to thank the German Research Foundation (DFG) for supporting this work with grant number 496307198. Furthermore, the authors want to thank Tim Lukas Kessel and Malte Nilges (both Technical University of Darmstadt) for their help and support in the development of the printed circuit board (PCB) for testing. Large language models were used to improve the language of the manuscript. Grammarly by Grammarly, Inc. was used for spell checking. ChatGPT by OpenAI and DeepL by Deepl SE were used to enhance the clarity of the writing.

## Author contributions

M.G. conducted the circuit design, simulation, and layout for the manufactured chip. M.G. built the test setup, ran the experiments, and was the primary author of the manuscript. K.H. supervised the work, provided guidance, and contributed to the writing and revision process. All authors reviewed the manuscript.

## Funding

Open Open Access funding enabled and organized by Projekt DEAL.

## Competing interests

The authors declare no competing interests.
