## [Peer Review File · Communications Engineering]

Reviewers' comments:

Reviewer #1 (Remarks to the Author):

Authors present mixed-signal design implementation for Oscillatory Ising Machines to solve optimisation problems. The current work contributes to the state of the art by providing a CMOS 28nm chip design which is competitive in design implementation and number of oscillators and coupling elements. This paper presents a tremendous engineering effort to implement such a large OIM. The paper should be accepted, though several points should be addressed:

1 - oscillator choice and design, require more description. It seems that a differential oscillator is chosen but not sufficient details provided

2 - the added adjustable delay line and calibration, should be described a bit better to grasp its necessity. Also, wouldn't such added delays corrupt the intrinsic dynamics of coupled oscillators and detriment the recurrency of the network.

3 - benchmarks are generated and it is not clear why not open source benchmarks are utilised for comparison as in previous works. Can other benchmarks such as tsp or be embedded and experimentally checked? How reconfigurable is the architecture?

4- weighted graphs and reprogrammability of the weights should be explained more clearly both on implementation and experimentally (Fig 3). It seems solutions gets better with weighted graphs.

5 - Delay tuning and calibration is performed at the start or during runtime? Accordingly it can impact the outcome and also runtime.

Reviewer #2 (Remarks to the Author):

The manuscript presents original work in the field of Oscillator-based Ising Machines, with a test chip fabricated entirely in 28nm CMOS technology. The test chip is made to solve randomly-generated graph partitioning problems, the authors reporting on solution quality, speed and energy-efficiency against various adjustable circuit parameters. This topic has garnered some interest in recent years, with prior publications that are well referenced by the authors. In general, the text is well written and provides a clear and accurate description of the work. This study appears to be on par with the state-of-the-art on the common metrics, and additionally, demonstrates experimentally several novel features that are of interest to the engineering community. Specifically, the implementation of a non-local coupling scheme as well as practical solutions for frequency, phase and delay calibration. The coupling weights resolution is quite advanced, and the network supports the application of individual biases. The network size, 1440 oscillators, is among the largest reported for this type of network.

In my opinion, the originality, novelty and technical value are significant enough for this work to be considered for eventual publication in Communications Engineering. However, the manuscript in its current form does have some flaws and blind spots, which I think must be addressed beforehand by mandatory revisions.

My main remarks are the following:

- The abstract claims that the routable connection capability simplifies the time-consuming graph embedding procedure. This argument can be understood qualitatively, as embedding an arbitrary graph does tend to be simpler on average as the number of allowable connections increases. However, the claim is not supported by any quantitative data, which deprives the readers of gaining some perspective on the added value of this feature. In fact, the non-local coupling strategy implemented in this work only enables increasing the maximal graph connectivity from ~0.9% to ~1.1% (more than 1M edges for all-to-all connectivity, 11724 implemented here, among which 9140 are local). Of course, those percentages would vary with network size. I am not so sure that the time savings when embedding very dense graphs are really significant, and I think either supporting data or a discussion are needed there.

- The authors argue in the introduction that all-to-all connected graphs are "unfeasible for large networks" due to the quadratic scaling of the number of connections. In other works dealing with analog in-memory computing, crossbars of conductive elements are often proposed to implement fully-connected layers in ANNs, or even restricted Boltzmann machines. It is not very clear to me

why this solution would be infeasible, and I think it would be useful if the authors could elaborate. Does this comment specifically apply to OIM because phase-encoding is so sensitive to parasitic delays ?

- I appreciate the efforts by the authors to benchmark their chip by claiming a computing speed and energy in the abstract. However, time to solution and therefore energy should strongly depend on problem complexity and targeted optimality. While some problems may be very difficult to solve with 100% guaranteed optimality, they could also possess a large number of easy-to-find near-optimal solutions. For example, I am not sure anybody could tell how computationally easy or hard it would be to reach 95% optimality on a graph partitioning problem with 0.5% of non-zero coupling edges. How many conflicts, or local minima are there in the cost function ? How far from optimal are the local minima ? In this sense, comparing the OIM to the time it take for Gurobi to give a solution with guaranteed optimality could also be misleading. Thus, my recommendations are the following: rather using time-normalized metrics such as power, being more transparent regarding the density of the problems solved by the OIM chip, comparing software and OIM performance at equivalent optimality.

- The main focus in the paper is admittedly area efficiency and compensation of device variability. One crucial aspect that I do not think is stressed enough in the manuscript is how to make sure that the network is operating in the proper balance of phase binarization versus coupling strength. The authors explain that the SHIL signal is gradually increased during the procedure to "gently freeze and discretize the oscillator phases". This raises a couple of important questions which should definitely be commented on:

○ Is there any useful computation occurring before the phases have been properly binarized by SHIL ? How is the computational space defined before then ? How can one be sure that the problem is correctly expressed if the nodes of the graph are not binary variables ?

○ Conversely, is it possible that past a certain point, the SHIL perturbation would become too strong and effectively screen the mutual coupling perturbations encoding the problem itself ? Could that "overflow" result in a degradation of the solution quality ?

○ Assuming a maximum was defined for SHIL strength, is there any impact of ramping speed on the solution quality (how "gentle" does it have to be and why) ? Basically, did the authors have to select min/max/rate values for the ramp, and how important are they?

- On a related topic, I am not very sure what the authors mean by the breakdown of the computation time, i.e. 200ns for coupling, 182ns for SHIL and 568ns for measurement. I think it should be clearer. What happens during the first 200ns ? Do the subsequent 182ns correspond to SHIL ramping time ? Does the "measurement" step include the stabilization time ?

- "Since OIM computing is non-deterministic due to noise, meaning that the computed solution varies randomly when computing the same problem again": I think this statement is controversial and potentially incorrect. For all we know, the noise could be negligible compared to the coupling and SHIL perturbations. As a matter of fact, the computed solution can vary randomly despite a deterministic computing process, for example if the initial states vary randomly. The authors later acknowledge that they have no control over the initial phases at the start of the computation, and that is in fact a likely explanation for obtaining different results across several runs. The claim is hasty at best, since the manuscript does not include any characterization of the noise that the authors refer to.

- On max-cut encoding: "The weights are limited to positive only values for better comparability of the obtained cut." I am confused by the sign, I would expect max-cut to feature negative weights. For graph partitioning, we want as many $s_i s_j$ products as possible to be negative. In the Ising Hamiltonian eq. (1), the energy function is minimized if the corresponding J_{ij} are also negative.

Finally, please find listed below some more minor remarks:

- The following excerpt in the abstract: "Despite continuous improvements in digital processors, solving non-deterministic polynomial-time (NP)-hard optimization problems with classical algorithms results in exponentially growing runtime as the problem size increases. Oscillator-based Ising machines (OIMs) offer a promising alternative by exploiting the analog coupling between electrical oscillators to efficiently solve such optimization problems." is ambiguous. Although I do not presume that it was the authors' intent, one could read this and think that OIM execute non-classical algorithms that can solve optimization problems in less than exponential time. Such a claim would be inaccurate (or at least, unsubstantiated). Solving optimization problems by local search algorithms over binary variables that resemble the Ising model is not a new concept. However, one could hope that adapting the hardware (components, dynamics, topology etc.) to the solving approach would indeed yield some benefits in terms of speed and/or energy consumption.

- In the introduction: " J_{ij} (...) is the matrix with coefficients of the interaction between spins". Strictly speaking, J_{ij} is not a matrix, but rather its coefficients.

- There is probably a clearer way to illustrate the conversion of Ising to QUBO than Eq.(3). Eq.(3) is not under its simplest form (Q_{ij} term remains on both sides), and only expresses the quadratic terms $x_i x_j$. I think merely indicating the variable change $x_i = (1+s_i)/2$ would be easier to understand and to use. It would also immediately follow that $x_i x_j = [(1+s_i)(1+s_j)]/4$.

- "The concept of computation using OIMs is outlined in Fig. 1a. The actual optimization problem must first be formulated in an Ising or QUBO form." As pointed out by the authors just before, wouldn't phase-encoded OIM naturally map to Ising models, and not so much to QUBO? Wouldn't a QUBO-to-Ising transformation need to take place before using the OIM?

- Fig. 3(a) x-axis label is clipped

- "As any arbitrary oscillators can be connected through the routing network on demand, we significantly simplify the embedding process making our system well-suited for sparse problems." I find this claim particularly vague (how sparse, how much is significantly, simpler embedding compared to what?) and I did not see data supporting it. My understanding is that although some non-local coupling is enabled by the routing network, there are still connectivity constraints. And in the following, the authors do not perform graph embedding but rather solve randomly assign coupling weights within the connectivity constraints of the chip.

- A comment would be welcome regarding how the sign of the bias terms h_i is handled. Is the reference clock arbitrarily set to the +1 (or -1) phase state?

- I would suggest using different reference names as column headers in Table 1, maybe "F. Author et al." rather than the name of the conference or journal.

Reviewer #3 (Remarks to the Author):

See attached file.

Review file for “An Integrated Coupled Oscillator Network to Solve Optimization Problems” Graber & Hofmann

In this work, the authors propose, fabricate, characterize and benchmark a CMOS-based oscillatory Ising machine with a new semi-reconfigurable coupling scheme.

The most promising Ising machines are those that offer the greatest connectivity between spins because it reduces the complexity of the embedding step - eg mapping the graph of the problem to be solved to the graph of the Ising machine. As the authors highlight in the introduction, this embedding step can both be also a NP-hard problem, deteriorate the solution of a problem given by the Ising machine because of the chain breaking issue (when the multiple hardware spins used to represent a single logical spin have a different value at equilibrium) and can result in a ridiculously poor usage of the hardware spins.

Until now, most of the works propose hardware Ising machines with a fixed sparse connectivity graph. Only a few works propose either all-to-all couplings (I'll cite a few: optical coherent Ising machines with measurement-feedback schemes (McMahon et al., Science 2016), all-to-all coupling with CMOS-based oscillatory Ising machine (Lo et al., Nature electronics 2023)) or reconfigurable connectivity (Field-Programmable Ising Machines, Jagielski et al., arxiv:2306.01569)). This work is very much in line with recent developments in the field of Ising machines and is thus interesting.

My assessment is that this work is a nice piece and should receive great attention in the Ising machine community. However, I would be happy to recommend publication only if some corrections/clarifications are made.

Below I detail the main contributions of the work and detail the questions/ points that I would like the authors to address before potential publication.

Main contributions:

- To overcome the issue of fixed (and often sparse) connectivity between hardware spins, the authors propose to add on top of a local dense connectivity between clusters of spins cluster of 11 all-to-all connected spins), routable long-range connections between long-distant spins (up to 4 long-range coupling per spin if routing track between those spins is available).
- The spins are implemented as CMOS ring-oscillators where the spin value is encoded in the phase of the oscillator, enforced to project on either 0 or pi phase with respect to a reference clock through a standard second harmonic injection locking scheme.
- To realize this scheme with routable long-range couplings, the authors arrange the densely connected clusters of spins in between routing buses. Each spin is connected to the routing tracks by two ports: one for sending its current phase value which is sent to another distant spin to which it is coupled to, the second port to receive phase information from the distant spins to which it is coupled to. Locally, the spin block converts the input digital phase information into analog currents that are summed and sent back to the ring oscillator to modulate its phase (as well as current from neighboring spins).
- The authors designed and fabricated both the ASIC that implements that specific scheme as well as the testing hardware, which is impressive.
- The authors benchmark their chips on different MaxCut benchmark tasks and compare the results they got with their chip with the ground truth solution (apparently found through a brute-force search approach). Their chip compares favorably in term of probability to find a solution close to the ground truth.
- Furthermore, the authors show how different parameters (coupling strength, frequency of the oscillators) affect the performance of the chip for solving MaxCut problems.

Questions/ points:

- In the paper, no mention is made of the « Field programmable Ising machine » (FPIM) (Jagielski et al., arxiv:2306.01569) while it seems the closest work in the literature to date to this work. I strongly encourage the authors to
 1. Cite this work in introduction to position their work with respect to the existing literature and
 2. Add a paragraph in the discussion to again discuss the pros/ cons of each approach. Also, why in the FPIM paper no mention is made to the delays for the couplings between distant spins? As the authors show here, the delay can alter the solution of problem. Maybe this is because the problems studied here require long-range connections while the SATs problems studied in the « FPIM» paper do not? (See my point below)
- While the routable connectivity is very interesting, the truth is that it « only » allows to add 4 (4 given page 3 - but is it 5? - in Table 1, 11 + 4 adds to 16 apparently? Is there a typo?) long-range couplings to the spins. This limitation was not written explicitly in the abstract neither in the introduction. Could the author write this explicitly? And could the authors discuss a potential way to scale up the number of long-range couplings per spin? (If there exist one?)
- It's not clear how the problems graphs are embedded on the chip. For instance, page 4 it is written «*The routing enables the connection of two arbitrary oscillators on the chip together if suitable routing tracks are not already occupied.* » which is not clear at all. How the spins are prioritized over others on a specific routing track? Random choice of the spins? Sequential choice? The authors should better describe the embedding technique.
- Figure 2: Overall Fig. 2 is very clear however there are several acronyms that are used and described only later in the main text. Could the authors add a description of the acronyms in the caption of the figure too? It will help the reader to understand the figure quicker.
- Figure 2: Fig. 2b could be a bit misleading in the sense that the reader could understand that a spin could be coupled to 7 long-distant spins through the routing (because of the 6 light gray couplers for routed signals) whereas it can only be coupled to 5 long-distant spins? (I might be wrong, and again value only guessed from Table 1). Similarly, the reader could understand that a spin is only coupled to 4 neighbor spins (for the same reason of 3 light gray couplers from local neighbors). For those reasons, the authors should update Fig. 2b.
- Page 4: what is the cost of having one DAC per parameter on the chip? It seems very inefficient in term of chip area - also programming the parameters could be done in time-multiplexed way and use only one DAC per node? I doubt that all DACs can be address in parallel during the parameters setting phase so it should not be too much time wasted.
- Page 4: the authors describe succinctly the method for overcoming delays caused by routing the signals across the chip. The method might need to be better explained in the methods section (I had to read their earlier paper on this specific method).
- Benchmarks part: This section is really not clear:
 - Let's take the problem tackled in Fig3.a. I assume the authors are solving the MaxCut problem with local connection between spins where the couplings are all similar. Then, how can one define 100 different problems while this problem is unique? ie solve the

- MaxCut with local coupling which value is 1. Then, it follows that it is really not clear what « best run » and « average run » means in the histograms. To me, best run should be an individual run and not represented as a histogram but rather as a single data point on the plot. Same for an average run, to me it should report the average performance as a single data point on the plot.
- For Fig3.b I understand more as we can create 100 different problem instances and we have 100 data points on the plot. However, it should be written clearer in the text how many problem instances are created, how they are created and how many repetitions are performed for each problem (it is written in the text but very not clear).
- The problems used for benchmarking the chip (MaxCut) are standard problems for Ising machines but they might not be the most relevant problems for such sparse programmable connectivity graph, and it's also not clear for which kind of problems this specific semi-reconfigurable connectivity would have an advantage.
 - Could the authors show/ discuss the absolute advantages of their approach with respect to a standard embedding procedure? In this version, it's not clear what is the absolute advantage of their approach with respect to other approaches used in other Ising machines (eg. Embedding or all-to-all connectivity).
 - It is crucial to show the time for creating the embedding versus their approach on a specific problem to show a potential advantage (and if to show an advantage one has to find a specific kind of problem this is fine!). Also, the authors could report the statistics of the solutions of their approach versus the statistics of other approaches using embedding (as a claim in the introduction is to get rid of embedding that can alter the solution of a problem!). The authors could use a python package (the DWave package for instance) to simulate the embedding of a particular problem on a specific graph (Chimera, Pegasus) and the resolution with Simulated Annealing and report the results to compare with their solution and time to embed the problem.
 - We want to see what the scheme/ chip really adds in value. The authors did a great job in Fig. 1a for clearly explaining the workflow for solving any problem on an Ising machine - so they could simply show numbers on the same chart of their approach versus other standard approaches (with sparse graphs + embedding).
 - SAT problems are also standard problems used to benchmark Ising machines and are often sparse. My opinion is that SAT problems could be more relevant to benchmark the chip with respect to other IMs that have a sparser connectivity and rely on embedding. The authors might want to add another benchmark with SAT problems that map directly to their chip connectivity to show the real advantage of their approach.

Minor comments:

- Page 1: "A part of bridging the gap between desired and available computing power are Ising machines »: the authors should cite the reference review paper on Ising machines here (Ising machines as hardware combinatorial solvers, Mohseni et al. Nature reviews physics, 2022)
- Page 1: it is written «Researchers are exploring different physical principles such as quantum mechanics³, optics^{4,5}, memristors⁶, and spintronics⁷». I understand the sentence, but while quantum mechanics could be somehow seen as a principle, optics, memristors and spintronics are not physical principles but rather physical substrats that exploit different physical principle to compute (optics with coherent Ising machines: minimum loss, memristive array leverage the Kirchhoff laws for computing, etc). Maybe the author should rephrase this sentence.
- Page 2: it is written « The oscillators run and mutually manipulate their phases »: maybe the authors could find a better word than « manipulate » in this sentence (influence or modulate are the first words that come to me, but another one could be more appropriate?). Manipulate is also used 2/3 times later in the main text.
- Page 2: just after, it's not clear that the authors threshold the phase values to get spins values after measurement (while it's clear from Figure 1). The authors should explicitly state that in the text.

- Page 2: it's not clear how the letter G was realized on the chip: only individual biases applied to the spins? No spin-spin couplings? The authors should better describe in the text (and/ or figure caption) how the letter G was realized.
- Page 2: "the graph embedding of the problem graph into a hardware network can be NP-hard as well » : maybe the word « connectivity graph » is to be preferred to "network ».
- Figure 2: a schematic of the dense local connectivity could help the reader to visualize the actual connectivity - it's described quickly in the text (page 3) but it's not very clear.
- Figure 3: Fig3.c is overlapping on Fig3.a (especially the x label)
- Page 5: "The parameters like the coupling strength were individually tuned for each problem set. » - It's not clear how the parameters were tuned - to optimize which metric? Is it only related to Fig4.a?
- Page 5: how can the authors sweep over the coupling value for the « weighted » problem as for this problem, each eight has a different value? (Over the available range of accessible value for the couplings)
- Can the benefit of operating at a lower frequency the oscillators be linked with the delay of propagating the spin value to distant spins? If the frequency is lower, then the period of an oscillation is longer which makes the delay less a problem?
- Given the locally dense connectivity the chips offer with long-range routable couplings, could the authors think of better problems than MaxCut or SAT to be solved on this IM? (I am thinking of solving more layout-driven tasks such as the convolutional neural network trained on an Ising machine (Laydevant et al., arxiv:2305.18321, 2023) but not only this specific example).
- I have a last question, which is more a curiosity question from my side, about the Second Harmonic Injection Lockin (SHIL) used for reading out the phase of the oscillators. As the authors wrote in the manuscript, the SHIL schedule is somehow an equivalent to a temperature schedule used in simulated annealing. Can the authors comment on the choice on the SHIL schedule they have used? Have the authors tried different schedule similar to commonly used temperature schedule? ($1/\sqrt{T}$, $1/(1+\log(T))$, etc)? It's not clear which SHIL schedule has been used while it is written « The SHIM-injector gradually applies the perturbations for the SHIL... » page 4.

Dear Reviewers,

Thank you very much for the detailed comments. We appreciate your time, dedication, and attention to detail. This is one of the best feedback we've ever received for a paper. It helped us very much to improve our manuscript, which will hopefully contribute to the research community.

We've added a response about the changes in the manuscript to every of your comments (in green). If applicable, written changes in the manuscript are marked in (in blue) for convenience. At some comments, we added some background information as a response, since it is too detailed to include in the manuscript. In case there are still open comments/questions, that should be addressed, we are happy to help.

As with probably every topic, there are way too many details to discuss everything in a paper. In the case of our OIMs, the level of interdisciplinary ranges starts on a circuit design level and touches the (mathematical) application level at the top. We concentrate our findings on the chip level and the experimental results of our prototype.

We noticed that most comments, which we could not fully incorporate in our manuscript, are related to the embedding. Unfortunately, the embedding of optimization problems into a hardware graph is in itself already a very complex topic with many publications. While we would like to provide more comprehensive benchmarking with open-source problems and other problem types as suggested, we are lacking a capable embedding algorithm/flow to solve arbitrary problems on our machine.

We have done previous work in the area of embedding for OIMs (doi: 10.1109/Austrochip56145.2022.9940722), which greatly motivated our routable design. Unfortunately, despite further improvements in the meantime, our embedding heuristic is not capable enough to successfully embed problems of our current system size. To our surprise, embedding is actually much more difficult than initially expected. We do not have the resources to develop a custom competitive embedding, which would be necessary for a fair comparison on a whole system level. Companies like D'Wave put serious effort into the software stack to support their hardware. However, we do not think that this reduces the impact of our paper too much. Using hardware-specific generated problems (as other groups do as well, e.g. 10.1038/s41928-022-00749-3) one can avoid this complicated embedding, which is sufficient for evaluation of the chip.

Your sincerely,

Klaus Hofmann und Markus Graber

Reviewer #1

Authors present mixed-signal design implementation for Oscillatory Ising Machines to solve optimisation problems. The current work contributes to the state of the art by providing a CMOS 28nm chip design which is competitive in design implementation and number of oscillators and coupling elements. This papers presents a tremendous engineering effort to implement such a large OIM. The paper should be accepted, though several points should be addressed:

1 - oscillator choice and design, require more description. It seems that a differential oscillator is chosen but not sufficient details provided

Details added:

“We use a differential delay cell design shown in Fig. 2d, which was empirically optimized for its high sensitivity to coupling currents. The small number of 4 delay cells provides a good sensitivity but still sustains oscillation at strong injected currents reliably. The differential architecture provides phase shifts of π by swapping both differential signals. This is used to implement in-phase (0-phase difference) and anti-phase (π -phase difference) coupling. Additionally, the higher power supply rejection ratio (PSRR) of differential designs reduces unwanted coupling via the power supply”

2 - the added adjustable delay line and calibration, should be described a bit better to grasp its necessity. Also, wouldn't such added delays corrupt the intrinsic dynamics of coupled oscillators and detriment the recurrency of the network.

Based on the comments of another reviewer, we added the section “Delay-Line Calibration” in methods. Additionally, we emphasized this in the manuscript:

“The received time-delayed signal creates a perceived phase lag at the distant oscillator, that does not match the actual phase of the receiver. When coupled, the oscillators will react based on this perceived phase potentially causing a runaway situation and a frequency shift.” [To address this, we have introduced a delay compensation scheme, the design and simulation of which are discussed in our previous work [citation]. This method essentially transmits the oscillator signal of the previous stage, which can be seen as the oscillator waveform with a "negative" delay.

An adjustable delayline supplements the propagation delay through the routing network to match that "negative" delay so that the received oscillator waveform is time synchronous with the actual transmitting oscillator.]“ So, the perceived phase at the receiving oscillator matches the phase of the transmitter. However, the phase dynamics of the coupling are altered, because any response is perceived at the receiver with a full period delay.”

The intrinsic and unavoidable propagation delay changes the phase dynamics. If this gets too large (roughly $> 10\%$ of the oscillator period, although this is not a hard limit) coupling is severely affected. There are two ways to deal with this: A) Reduce the operating frequency so that the signal propagation delay gets small; B) find a way to deal with that delay (e.g. our compensation scheme). Option A is straightforward but can lead to severely reduced speed (Think of a very large chip with e.g. 100mm² and the delay when routing through potentially thousands of switch blocks). Out of research interest, we are hence focusing on B. As cited, we dedicated a paper on simulation/design of the routing: (10.1109/ICECS58634.2023.10382841). Nevertheless, this approach affects the phase dynamics and is thus not perfect, which can be clearly seen from the measurements here. We are not sure if something like a delay compensation without changing the phase dynamics can be physically even possible. More research is needed in this direction and as far as we know, we are presenting a first approach on that. There is very interesting work on coupling PLLs in the presence of large delays (e.g. 10.1109/TCSII.2022.3176827). However, we do not see how this can be applied to the fast coupling of OIMs.

3 - benchmarks are generated and it is not clear why not open source benchmarks are utilised for comparison as in previous works. Can other benchmarks such as tsp or be embedded and experimentally checked? How reconfigurable is the architecture?

Unfortunately, most open-source benchmark sets are for/from the optimization community in mathematics. They require more nodes and/or more connections per node than our hardware system can provide. For example, only G11, G12, and G13 of the popular G-set could even fit into our system based on the number of edges and oscillators. Additionally, the embedding is quite a difficult task. (We have done work on that, but it is not powerful enough to embed problems with our system size)

We wish we could solve open-source benchmarks for better comparability, but it is just not possible. Hence, we published our benchmark set on figshare (see Data availability), which is tailored for our system to avoid the embedding. Other groups did similar things (e.g. [10.1038/s41928-022-00749-3](https://doi.org/10.1038/s41928-022-00749-3))

Together with comment 4, we added a dedicated section “Chip Reconfigurability & External Adjustments” to explain the reconfigurability.

4- weighted graphs and reprogrammability of the weights should be explained more clearly both on implementation and experimentally (Fig 3). It seems solutions gets better with weighted graphs.

We added a dedicated section “Chip Reconfigurability & External Adjustments” explaining all configuration options of our chips, which are used in the manuscript. For the interested reader, this should clarify, how we configured our chip.

Yes, the weighted problems show slightly less spread and are marginally better than the difficult set, which uses weights of the same strength. We are not yet sure why exactly this is the case. However, we have a strong suspicion. Due to the varying weights, the weighted set has fewer “conflicts”, where multiple oscillators are coupled with the same strength in impossible configurations. E.g. a “conflict” is something like a ring of three anti-phase oscillators coupled with the same weight. There the coupled phases cannot be achieved and the same weights fight against each other.

5 - Delay tuning and calibration is performed at the start or during runtime? Accordingly it can impact the outcome and also runtime.

Added this to the methods as subsection “Calibrations”. In short: Always at the start and when the operating parameters are significantly changed.

Reviewer #2

The manuscript presents original work in the field of Oscillator-based Ising Machines, with a test chip fabricated entirely in 28nm CMOS technology. The test chip is made to solve randomly-generated graph partitioning problems, the authors reporting on solution quality, speed and energy-efficiency against various adjustable circuit parameters. This topic has garnered some interest in recent years, with prior publications that are well referenced by the authors. In general, the text is well written and provides a clear and accurate description of the work. This study appears to be on par with the state-of-the-art on the common metrics, and additionally, demonstrates experimentally several novel features that are of interest to the engineering community. Specifically, the implementation of a non-local coupling scheme as well as practical solutions for frequency, phase and delay calibration. The coupling weights resolution is quite advanced, and the network supports the application of individual

biases. The network size, 1440 oscillators, is among the largest reported for this type of network.

In my opinion, the originality, novelty and technical value are significant enough for this work to be considered for eventual publication in Communications Engineering. However, the manuscript in its current form does have some flaws and blind spots, which I think must be addressed beforehand by mandatory revisions.

My main remarks are the following:

- The abstract claims that the routable connection capability simplifies the time-consuming graph embedding procedure. This argument can be understood qualitatively, as embedding an arbitrary graph does tend to be simpler on average as the number of allowable connections increases. However, the claim is not supported by any quantitative data, which deprives the readers of gaining some perspective on the added value of this feature. In fact, the non-local coupling strategy implemented in this work only enables increasing the maximal graph connectivity from ~0.9% to ~1.1% (more than 1M edges for all-to-all connectivity, 11724 implemented here, among which 9140 are local). Of course, those percentages would vary with network size. I am not so sure that the time savings when embedding very dense graphs are really significant, and I think either supporting data or a discussion are needed there.

Unfortunately, we do not have meaningful quantitative data to present in this manuscript. The idea is driven by our earlier work on embedding heuristics, where we concluded, that routing improves the success rate significantly (e.g. 10.1109/Austrochip56145.2022.9940722, Fig. 7). However, that presented heuristic is not capable enough for the current system size and needs massive improvements.

In our experience, it is really difficult to embed every edge into the hardware graph. When sequential mapping the edges, it gets increasingly difficult the more edges are already mapped. With more mapped edges, the choices are reduced, so that usually our embedding algorithm fails because it cannot embed the last few connections (Actually there are a lot of reorganization and re-mapping before the algorithm will eventually give up after too many iterations). That is the reason why we the routing channels can make the embedding much easier. The last remaining connections, which can not be embedded using the local connectivity would usually require a significant reorganization of the existing embedding. With the routing channels, the embedding algorithm can use these for the remaining, hardest connections of the embedding.

So, we improved the manuscript with an explanation:

“Hence, we propose a configurable, routable network with strong local connectivity as a trade-off. The idea is, that most of the graph edges should be embedded into the local connectivity. The routing network is just used for the remaining edges, which cannot be embedded locally. While the first edges can usually be easily mapped to the hardware connectivity graph, it gets increasingly difficult to find valid mappings for the remaining edges the more nodes and edges are already mapped to a hardware component. By using the flexible routing connections on demand for these remaining challenging connections, the difficulty can be reduced. Hence, the addition of such routable channels simplifies the embedding and can increase the success rate compared to a local network [citation: 10.1109/Austrochip56145.2022.9940722].”

While we totally understand, that our system is very sparse with roughly 1% connectivity, the focus is more a demonstration of the concept including routable channels. We want to show and evaluate if routing is a feasible option and how such systems perform with a decent size of nodes.

- The authors argue in the introduction that all-to-all connected graphs are "unfeasible for large networks" due to the quadratic scaling of the number of connections. In other works dealing with analog in-memory computing, crossbars of conductive elements are often proposed to implement fully-connected layers in ANNs, or even restricted Boltzmann machines. It is not very clear to me why this solution would be infeasible, and I think it would be useful if the authors could elaborate. Does this comment specifically apply to OIM because phase-encoding is so sensitive to parasitic delays ?

To make it more clear we changed this to: "the amount of connections $n/2 * (n-1)$ scales quadratically with the network size n , making it challenging for large networks. Besides the immense area, the coupling can get difficult. For example, the delay and capacitive load introduced by the numerous connections poses a serious challenge."

As far as we are aware, all-to-all connections were reported to be with 16, 30, and 48 nodes. The 48-node system (10.1038/s41928-023-01021-y) has actually 100 all-to-all connected on their chip (arranged as four 50 all-to-all blocks). However, they reported issues that the oscillators did not lock reliably for the 100 size). The 48-node implementation actually uses a crossbar-like implementation as suggested. A 1000 node system needs already 100x more connections and therefore more than 100x the area of the 48 system (respectively 100). Sparse networks are available up to 1968 Nodes (10.1038/s41928-022-00749-3).

One challenge is the area demand with all-to-all connections. The phases need to be transmitted simultaneously to all other nodes without too much delay. The other challenge is the capacitance/load. Making a physical connection will introduce a massive capacitance, which severely affects the oscillator.

- I appreciate the efforts by the authors to benchmark their chip by claiming a computing speed and energy in the abstract. However, time to solution and therefore energy should strongly depend on problem complexity and targeted optimality. While some problems may be very difficult to solve with 100% guaranteed optimality, they could also possess a large number of easy-to-find near-optimal solutions. For example, I am not sure anybody could tell how computationally easy or hard it would be to reach 95% optimality on a graph partitioning problem with 0.5% of non-zero coupling edges. How many conflicts, or local minima are there in the cost function ? How far from optimal are the local minima ? In this sense, comparing the OIM to the time it take for Gurobi to give a solution with guaranteed optimality could also be misleading. Thus, my recommendations are the following: rather using time-normalized metrics such as power, being more transparent regarding the density of the problems solved by the OIM chip, comparing software and OIM performance at equivalent optimality.

We want to clarify, that we did not compare the time of Gurobi with our chip. In the manuscript, we have written: "To underline the higher difficulty, computing the reference solution using the commercial software Gurobi[40] for the **difficult set** took on average 10,000x longer than the **simple set**." So, we used the runtime of Gurobi (on the same system) to have some form of judgement, that there is quite a substantial difference in difficulty between our used benchmark sets.

The provided absolute numbers in the abstract are intended to potentially give some sort of comparison with other comparable (OIM) systems, which is later provided in Table 1. We did not intend to suggest a direct comparison with other systems. However, energy and time savings are the natural motivation for research in Ising machines. As you point out the power varies with the problem. In the manuscript, we already provide more data on the power consumption: "The energy

consumption per computed problem varies between 404 nJ and 466 nJ across the individual problems of the later discussed *difficult* benchmark set. A single oscillator including its periphery has a measured power consumption of 113.3 μ W and a coupling connection needs approximately 23 μ W." We think, that this is already very transparent regarding the density of the problems solved.

We totally agree that it is difficult to do a meaningful comparison between different computing approaches. All feasible benchmarking options we can think of are some kind of "comparing apples and oranges" situation. Using an exact solver such as Gurobi cannot be a fair comparison to a 95% heuristic. The Time/Energy of a whole computing system including network, storage, operating system, power supply with a single chip is also not a fair comparison.

How would a heuristic or Gurobi know that it reached 95%? What is the stopping criterion? Something like the Goemans and Williamson algorithm, which guarantees 87.8% is the only option here. Overall, we are of the opinion, that the OIMs in their current research phase are too immature to provide a fair comparison. Larger implementations capable to compete with "real-world" benchmarks would be needed.

We appreciate the suggestion, however, they do not solve the raised questions in our opinion. We disagree that time-normalized metrics such as power is a better comparison than energy. A system consuming 1/10 of the power but needing 10x longer is not better.

- The main focus in the paper is admittedly area efficiency and compensation of device variability. One crucial aspect that I do not think is stressed enough in the manuscript is how to make sure that the network is operating in the proper balance of phase binarization versus coupling strength. The authors explain that the SHIL signal is gradually increased during the procedure to "gently freeze and discretize the oscillator phases". This raises a couple of important questions which should definitely be commented on:

- o Is there any useful computation occurring before the phases have been properly binarized by SHIL ? How is the computational space defined before then ? How can one be sure that the problem is correctly expressed if the nodes of the graph are not binary variables ?

This is part of the basic OIM operation, which was pioneered and discussed by the group of Jaijeet Roychowdhury and especially his Ph.D. student Tianshi Wang (e.g. doi: 10.1007/s11047-021-09845-3 (section 3.2 and 3.3), doi: 10.48550/arXiv.1709.08102, Ph.D. thesis of Tianshi Wang <https://www2.eecs.berkeley.edu/Pubs/TechRpts/2020/EECS-2020-12.html>). Hence, it does not need to be discussed in our manuscript, but we now added those references in the introduction when describing OIMs. In the scope of this and the related comments, we extended the description of the SHIL section (see the last of the SHIL comments).

Anyhow, the short answer is yes, it is already doing an approximate computing of the Ising Hamiltonian. However, the SHIL is crucial to ensure that the OIMs actually strive towards the minimum of the Ising Hamiltonian.

- o Conversely, is it possible that past a certain point, the SHIL perturbation would become too strong and effectively screen the mutual coupling perturbations encoding the problem itself ? Could that "overflow" result in a degradation of the solution quality ?

We did experiments on this but did not include them in the manuscript for space reasons. We think the other findings are more interesting/relevant.

FIGURE REDACTED: This information was removed due to sensitive data for a future publication

Essentially, the SHIL locking must overcome the coupling strength between oscillators to ensure locking. Without, the performance is very poor as the continuous phases of the oscillators are just rounded into the discrete state. When properly locked a good performance is reached. Beyond $9\mu\text{A}$, there is a tiny drop-off in performance. We've seen very similar behavior in our previous chip generation. This can apply even stronger SHIL (injection approx. 2x the bias current), which shows a more perceivable drop-off for very high strengths. Hence, the presented current generation has a reduced maximum SHIL strength. So, there is a "too much" (the injection current is already exceeding the bias current of the stage), but the choice is not as critical as e.g. the coupling strength.

For those results, it should be considered, that the SHIL strength affects the step size of the ramping (see answer to your next comment).

o Assuming a maximum was defined for SHIL strength, is there any impact of ramping speed on the solution quality (how "gentle" does it have to be and why) ? Basically, did the authors have to select min/max/rate values for the ramp, and how important are they?

Yes, our first-generation chip had an adjustable ramp-up and ramp-down of the SHIL which had adjustable strength as well. Based on experimental analysis of those, we selected a ramp-up in 8 steps, where each step takes a nominal oscillation period. Slower ramps did not improve the results and led to even worse results for longer durations (32 cycles and more). The ramp-down of the SHIL was dropped completely since it did not show any impact on the computing. The original idea of ramping down was, that the system computes the same problem again by slowly releasing it from the previous solution as starting point.

To address all the comments regarding the ramp-up, we extended this in our manuscript:

The SHIL-injector gradually applies the perturbing SHIL signal to gently freeze the oscillator phase and therefore discretize the oscillator phases in the two spin groups. This ensures, that the coupled oscillators try to minimize the Ising Hamiltonian [citation: Wang2021]. Based on the experimental evaluation of our previous generation with adjustable ramps, we decided to ramp up the SHIL in 8 linear steps, which are increased after one nominal oscillation period.

- On a related topic, I am not very sure what the authors mean by the breakdown of the computation time, i.e. 200ns for coupling, 182ns for SHIL and 568ns for measurement. I think it should be clearer. What happens during the first 200ns ? Do the subsequent 182ns correspond to SHIL ramping time ? Does the "measurement" step include the stabilization time ?

We refined it and it should be now clear:

“A single computation takes in total 950 ns. The first 200 ns are used to let the oscillators couple. Then the SHIL strength is ramped up to gradually force the coupled oscillators into the two phase groups, which takes 182 ns including a small margin for stabilization. Lastly, the phase measurement to obtain the computed result takes 568 ns.”

- "Since OIM computing is non-deterministic due to noise, meaning that the computed solution varies randomly when computing the same problem again": I think this statement is controversial and potentially incorrect. For all we know, the noise could be negligible compared to the coupling and SHIL perturbations. As a matter of fact, the computed solution can vary randomly despite a deterministic computing process, for example if the initial states vary randomly. The authors later acknowledge that they have no control over the initial phases at the start of the computation, and that is in fact a likely explanation for obtaining different results across several runs. The claim is hasty at best, since the manuscript does not include any characterization of the noise that the authors refer to.

Actually, this sentence is the reasoning, why we compute the same problem multiple times in our benchmarking. As we both agree, the outcome of an OIM computation can be non-deterministic. For example, we want to mention our previous generation: doi: 10.1109/ISCAS46773.2023.10181365 Figure 4, which clearly shows the varying outcome of multiple computations of the same problem.

First, I want to elaborate on the reasoning behind our claim. It is an analog computing principle and noise is the non-deterministic influencing factor, leading to two principles: A) The initial states are affected by noise. Especially the “lower” frequency components cause a slight drift of the oscillator frequency and thus initial phases of the oscillators. As we have written, there is no autocorrelation of the oscillator phase after just free-running for 1200ns (approx.. 120 periods) B) There is some noise affecting the coupling and SHIL process.

While we can neither confirm nor deny that “noise could be negligible compared to the coupling and SHIL perturbations”, we have still the strong opinion that this can affect the computing: When coupled without SHIL injection an oscillator can have any continuous phases. Then an oscillator could be in its continuous phase exactly between the two phase groups, which later determines the spin under SHIL injection. This can be likely, if e.g. the Hamiltonian will not change no matter in which state the corresponding spin is. Then the phase is in a sort of metastable state, which can be easily pushed in either of two states by even the smallest noise. As soon as they are SHIL locked, noise will likely have no influence anymore. When coupled, the impact of noise could be low as well. But we think that noise plays a crucial role during the transition from the coupled into the SHIL lock state for such metastable cases, which cannot be clearly assigned to one of the phase groups. Since OIMs do not have something like a clear noise margin, even the smallest noise could have an impact.

We see, that this is a controversial issue, one can extensively argue about. Due to the lack of control over the initial states, we can not provide any evidence if the computing starting from the same initial phase is deterministic or not. When thinking of the underlying differential equation without noise terms, the computation is deterministic given the initial state. When including any form of noise, we see a potential non-deterministic influence as discussed by a metastable state above. As this is just the reasoning for the repeated computations, we slightly changed the sentence:

“The outcome of an OIM computation using our chip depends on the random initial states and might be affected by noise, so that the computed solution varies randomly when computing the same problem again.”

- On max-cut encoding: "The weights are limited to positive only values for better comparability of the obtained cut." I am confused by the sign, I would expect max-cut to feature negative weights. For graph partitioning, we want as many $s_i \cdot s_j$ products as possible to be negative. In the Ising Hamiltonian eq. (1), the energy function is minimized if the corresponding J_{ij} are also negative.

The Ising model as shown in Eq. (1) has a negative sign before the sum term and the Hamiltonian should be minimized. Hence, positive coefficients J_{ij} in the Ising model should have positive $s_i \cdot s_j$ products (same spins) and negative coefficients J_{ij} in the Ising model should have negative $s_i \cdot s_j$ products (opposing spins).

For the partitioning in the maxcut, the cut is the sum of weights w , which span between the two sets. So, the nodes of edges with positive weight should be in different sets. The nodes of edges with negative weight should be in the same set, so that those are not included in the sum.

So, a positive weight of the maximum cut should be in different sets of the partitioning. For the Ising model this means, that the spins should be opposite, which is achieved with a negative weight. The transformation from the maximum cut to the Ising model is just inverting the sign of the weights (multiplication with -1). To make it more clear and avoid confusion, we changed this to:

"The edge weights of the maximum cut problem are limited to positive only values (transformed into negative J_{ij} coefficients of the Ising model) for better comparability of the obtained cut."

Finally, please find listed below some more minor remarks:

- The following excerpt in the abstract: "Despite continuous improvements in digital processors, solving non-deterministic polynomial-time (NP)-hard optimization problems with classical algorithms results in exponentially growing runtime as the problem size increases. Oscillator-based Ising machines (OIMs) offer a promising alternative by exploiting the analog coupling between electrical oscillators to efficiently solve such optimization problems." is ambiguous. Although I do not presume that it was the authors' intent, one could read this and think that OIM execute non-classical algorithms that can solve optimization problems in less than exponential time. Such a claim would be inaccurate (or at least, unsubstantiated). Solving optimization problems by local search algorithms over binary variables that resemble the Ising model is not a new concept. However, one could hope that adapting the hardware (components, dynamics, topology etc.) to the solving approach would indeed yield some benefits in terms of speed and/or energy consumption.

We do not intend to make any claim about the runtime complexity of OIMs. As far as we are aware, the time complexity of OIMs is not known. We can understand the potential ambiguity but see this as a lot of interpretation in what is written. Nevertheless, we changed this to avoid potential ambiguity:

"Despite continuous improvements in digital processors, solving NP-hard optimization problems with classical algorithms is very time and energy consuming. OIMs offer a promising alternative by exploiting the analog coupling between electrical oscillators to solve such optimization problems more efficiently."

- In the introduction: " J_{ij} (...) is the matrix with coefficients of the interaction between spins". Strictly speaking, J_{ij} is not a matrix, but rather its coefficients.

That's correct. We removed the "matrix" and just call it coefficient to avoid confusion to keep it simple.

- There is probably a clearer way to illustrate the conversion of Ising to QUBO than Eq.(3). Eq.(3) is not under its simplest form (Q_{ij} term remains on both sides), and only expresses the quadratic terms $x_i x_j$. I think merely indicating the variable change $x_i = (1+s_i)/2$ would be easier to understand and to use. It would also immediately follow that $x_i x_j = [(1+s_i)(1+s_j)]/4$.

I've used the multiplied form of $x_i x_j$, since it shows that a QUBO coefficient gets translated into a $s_i s_j$ term, a s_i and s_j term as well as a (not important) constant. This form emphasizes, that a Q_{ij} gets translated into a J_{ij} , h_i and h_j , where h_i and h_j are not always available in OIM implementations.

As a compromise, we change Eq. 3 to $x_i = (1+s_i)/2$ as suggested and just briefly mention the needed bias term h_i , h_j in the text. Since the transform is even available on Wikipedia (https://en.wikipedia.org/wiki/Quadratic_unconstrained_binary_optimization#Connection_to_Ising_models), we don't think the transform needs further elaboration anyways.

- "The concept of computation using OIMs is outlined in Fig. 1a. The actual optimization problem must first be formulated in an Ising or QUBO form." As pointed out by the authors just before, wouldn't phase-encoded OIM naturally map to Ising models, and not so much to QUBO ? Wouldn't a QUBO-to-Ising transformation need to take place before using the OIM ?

Our formulation is misleading here, sorry. The "actual optimization problem" in this context should be a problem originating from an application (e.g. optimizing a schedule/task/layout/portfolio/whatever). To make this clear, we change this to: "The concept of computation using OIMs is outlined in Fig. 1a. An optimization problem from an application must first be converted into an Ising form."

If a QUBO problem should be solved, it must be transformed into the Ising form and then brought to the chip. However, this is straightforward (see the previous comment) with a linear $O(n)$ runtime.

- Fig. 3(a) x-axis label is clipped

Fixed!

- "As any arbitrary oscillators can be connected through the routing network on demand, we significantly simplify the embedding process making our system well-suited for sparse problems." I find this claim particularly vague (how sparse, how much is significantly, simpler embedding compared to what ?) and I did not see data supporting it. My understanding is that although some non-local coupling is enabled by the routing network, there are still connectivity constraints. And in the following, the authors do not perform graph embedding but rather solve randomly assign coupling weights within the connectivity constraints of the chip.

Changed in the manuscript for clarity. See your first comment for applied changes.

This is correct, the claim is intentionally vague as the "difficultness" of the embedding is hard to grasp or even quantify. Compared to a system with the same local connectivity but without the additional routing network, the embedding is obviously "simplified". In our experience with the embedding algorithms (e.g. 10.1109/Austrochip56145.2022.9940722), we noticed, that the embedding often struggles because a few connections could not be established. It's similar to solving an optimization problem: Instead of embedding 100% of connections in the local network, it is sufficient to have e.g. 95% in the local connections and use the routing network for the remaining 5%, which could not be embedded in the local topology. So, the routing network drastically improves the chances. Unfortunately, our implementation is too inefficient to provide any embedding for a system of our size.

- A comment would be welcome regarding how the sign of the bias terms h_i is handled. Is the reference clock arbitrarily set to the +1 (or -1) phase state ?

We've added this to the explanation of the reference calibration:

"The distribution of the reference coupled oscillators, with one half coupled in-phase and the other half in anti-phase, is shown in Fig.5c (blue)." [...] "The phase threshold can be determined by finding the center of both reference phase(blue) groups. The phase angle of the +1 spin group can be clearly identified by the phase of the in-phase coupled oscillators."

More details about the phase threshold are included in the section "Phase threshold" in the methods parts.

- I would suggest using different reference names as column headers in Table 1, maybe "F. Author et al." rather than the name of the conference or journal.

There are no (project)names for each design. So, either journal/conference or author can be used. Since the citation is provided anyway, we do not see the choice as significant. However, we changed this based on the suggestion.

Reviewer #3

Review file for "An Integrated Coupled Oscillator Network to Solve Optimization Problems" Graber & Hofmann In this work, the authors propose, fabricate, characterize and benchmark a CMOS-based oscillatory Ising machine with a new semi-reconfigurable coupling scheme. The most promising Ising machines are those that offer the greatest connectivity between spins because it reduces the complexity of the embedding step - eg mapping the graph of the problem to be solved to the graph of the Ising machine. As the authors highlight in the introduction, this embedding step can both be also a NP-hard problem, deteriorate the solution of a problem given by the Ising machine because of the chain breaking issue (when the multiple hardware spins used to represent a single logical spin have a different value at equilibrium) and can result in a ridiculously poor usage of the hardware spins. Until now, most of the works propose hardware Ising machines with a fixed sparse connectivity graph. Only a few works propose either all-to-all couplings (I'll cite a few: optical coherent Ising machines with measurement-feedback schemes (McMahon et al., Science 2016), all-to-all coupling with CMOSbased oscillatory Ising machine (Lo et al., Nature electronics 2023)) or reconfigurable connectivity (Field-Programmable Ising Machines, Jagielski et al., arxiv:2306.01569)). This work is very much in line with recent developments in the field of Ising machines and is thus interesting.

My assessment is that this work is a nice piece and should receive great attention in the Ising machine community. However, I would be happy to recommend publication only if some corrections/clarifications are made. Below I detail the main contributions of the work and detail the questions/ points that I would like the authors to address before potential publication.

Main contributions:

- To overcome the issue of fixed (and often sparse) connectivity between hardware spins, the authors propose to add on top of a local dense connectivity between clusters of spins (cluster of 11 all-to-all connected spins), routable long-range connections between long-distant spins (up to 4 long-range coupling per spin if routing track between those spins is available).
- The spins are implemented as CMOS ring-oscillators where the spin value is encoded in the phase of the oscillator, enforced to project on either 0 or π phase with respect to a reference clock through a standard second harmonic injection locking scheme.
- To realize this scheme with routable long-range couplings, the authors arrange the densely connected clusters of spins in between routing buses. Each spin is connected to the routing tracks by two ports: one for sending its current phase value which is sent to another distant spin to which it is coupled to, the second port to receive phase information from the distant spins to which it is coupled to. Locally, the spin block converts the input digital phase information into analog currents that are summed and sent back to the ring oscillator to modulate its phase (as well as current from neighboring spins).
- The authors designed and fabricated both the ASIC that implements that specific scheme as well as the testing hardware, which is impressive.
- The authors benchmark their chips on different MaxCut benchmark tasks and compare the results they got with their chip with the ground truth solution (apparently found through a brute-force search approach). Their chip compares favorably in terms of probability to find a solution close to the ground truth.
- Furthermore, the authors show how different parameters (coupling strength, frequency of the oscillators) affect the performance of the chip for solving MaxCut problems.

Questions/ points:

• In the paper, no mention is made of the « Field programmable Ising machine » (FPIM) (Jagielski et al., arxiv:2306.01569) while it seems the closest work in the literature to date to this work. I strongly encourage the authors to

1. Cite this work in introduction to position their work with respect to the existing literature and
2. Add a paragraph in the discussion to again discuss the pros/ cons of each approach. Also, why in the FPIM paper no mention is made to the delays for the couplings between distant spins? As the authors show here, the delay can alter the solution of problem. Maybe this is because the problems studied here require long-range connections while the SATs problems studied in the « FPIM » paper do not? (See my point below)

We are aware of the pre-print of the FPIM paper but disagree that it is the closest work in literature. Jagielski et al. are working on an application/EDA level. They are motivated by the 3-SAT problem and modifies open-source FPGA tools to embed these problems into an Ising Machine. Hence, he is treating the analog oscillators just as a tile/box in his implementation. ("FPIMs use hardware spin representations (e.g., oscillators for OIM, ZIV diodes for BRIM, CMOS latches for BLIM, etc.) as primitive elements." as written by Jagielski et al.). Since they are not dealing with the actual analog implementation (as using oscillators in our case), they do not consider the analog implementation in detail. Consequently, there is no mention of the delay issue.

Their paper does not discuss the actual analog implementation and only mentions "A 1000-spin FPIM [...] would occupy about 10mm² in a 65nm process". They did a fabulous job of demonstrating a

program flow to solve SAT problems and Ising machines and analyzing the requirements for OIM implementation by evaluating possible embeddings.

We on the other hand are working on a hardware level and providing experimental evidence of its operation. We are much less working on an application level and focusing on advancing the analog implementation. So, we are working on an abstraction layer below Jagielski and would need his expertise to get requirements for our design to solve specific benchmark problems. The size of our system is mainly driven by the ease of implementation and should showcase the general capabilities of the computing including routing channels.

Consequently, we do not see how we could discuss the pros/cons of both of our approaches as Jagielski et al. and ourselves are on different layers. They are analyzing the requirements for a hardware connectivity network from an application (SAT problems) point of view, while we have done an actual hardware implementation without considering a certain application.

Interestingly, our chip implementation comes surprisingly close to what Jagielski et al. suggest. For solving the SAT-problem, he mentions that 8 tracks are needed. Our system has 12. For hardware reasons (implementing all possible connections/switches between tracks and nodes would massively increase the area), we have some restrictions (e.g. 4 connections per node/oscillator to these routing tracks, where every connection has a non-overlapping range of tracks it can access). Since the precise hardware connectivity is not explained in detail by Jagielski et al., we cannot judge if our system would fit their criteria. Nevertheless, we agree that his work should be mentioned:

“A very similar configurable topology is the “FPIM” (Field Programmable Ising Machine) proposed by Jagielski et al. [37]. They map boolean satisfiability problem (SAT) problems to a hardware representation and use a modified open-source field-programmable gate array (FPGA) tool flow to embed such problems in the hardware. This flow enables them to precisely analyze the needed hardware connectivity to embed a given set of optimization problems.”

A small side note: The idea of using routable connections similar to FPGAs is very obvious. For example, we mentioned this in our Paper in 2022 (doi: 10.1109/Austrochip56145.2022.9940722).

- While the routable connectivity is very interesting, the truth is that it « only » allows to add 4 (4 given page 3 - but is it 5? - in Table 1, 11 + 4 adds to 16 apparently? Is there a typo?) longrange couplings to the spins. This limitation was not written explicitly in the abstract neither in the introduction. Could the author write this explicitly? And could the authors discuss a potential way to scale up the number of long-range couplings per spin? (If there exist one?)

The connectivity is explained in the section “System”. The changes of Fig. 2b based on the comment about the light gray couplers make this also clear: 11 (local) + 4 (routing) + 1 (bias). For further clarity, we changed the entry in the table from “16” to “15+1” to account for the special role of the bias coefficient.

The architecture is designed to be flexible for scaling. One could just add (or remove) the desired couplers (including a DAC for it). This applies to local and routing connections. We expect that there are no other changes necessary when adding just a few couplers. At some point fine tuning (e.g. stronger output buffer, adapting summing circuit, changing strength, etc.) will be necessary:

Added after the description of the coupling summation circuit:

“In combination with the output buffer of the oscillator, which mitigates the capacitive loading caused by the connected couplers, several coupling connections can be simply added to achieve the desired connectivity.”

- It's not clear how the problems graphs are embedded on the chip. For instance, page 4 it is written «The routing enables the connection of two arbitrary oscillators on the chip together if suitable routing tracks are not already occupied. » which is not clear at all. How the spins are prioritized over others on a specific routing track? Random choice of the spins? Sequential choice? The authors should better describe the embedding technique.

Our current way to handle routing is added to the methods, sections “Path Routing”:

A more elaborate assignment of the spins to the physical network is part of the embedding algorithm, which is outside the scope of this work. Our own embedding heuristics are not capable enough and were not used in this work. Hence, we've been already planning to adapt parts of the VTR (Verilog-to-routing) flow (used by Jagielski et al. for their 3-sat). However, even this is not perfect for embedding as one could split or merge nodes/paths of the optimization problem. For example:

Fig. 3: Simplifications are applied to make mapping easier.

(10.1109/Austrochip56145.2022.9940722 Fig. 3)

- Figure 2: Overall Fig. 2 is very clear however there are several acronyms that are used and described only later in the main text. Could the authors add a description of the acronyms in the caption of the figure too? It will help the reader to understand the figure quicker.

Added in the caption as suggested.

- Figure 2: Fig. 2b could be a bit misleading in the sense that the reader could understand that a spin could be coupled to 7 long-distant spins through the routing (because of the 6 light gray couplers for routed signals) whereas it can only be coupled to 5 long-distant spins? (I might be wrong, and again value only guessed from Table 1). Similarly, the reader could understand that a spin is only coupled to 4 neighbor spins (for the same reason of 3 light gray couplers from local neighbors). For those reasons, the authors should update Fig. 2b.

Sorry, there was something mixed up with the light gray couplers. The figure is now improved and should be clear.

- Page 4: what is the cost of having one DAC per parameter on the chip? It seems very inefficient in term of chip area - also programming the parameters could be done in time-multiplexed way and use only one DAC per node? I doubt that all DACs can be address in parallel during the parameters setting phase so it should not be too much time wasted.

The DACs occupy approx. 12.8% of the whole OIM core area (= chip excluding I/O-Ring, I/O driver, etc.). They are designed to be as small as possible while offering 4-bit at a DNL < 1. Every DAC is essential just a binary weighted current mirror. All parameters are written to the chip with a simple serial interface. A high-performance network-on-chip (NoC) could be used in the future to maximize data throughput. However, the interface is not affecting the research on the actual OIM at all.

Time multiplexing is a very good idea, we added this briefly in the discussion: “A time-multiplexing of the DACs, which occupy 12.8% of the core area, could further save area.”

- Page 4: the authors describe succinctly the method for overcoming delays caused by routing the signals across the chip. The method might need to be better explained in the methods section (I had to read their earlier paper on this specific method).

Added section “Delay-Line Calibration”, including the figure of the cited paper which explains the principle. See comment 2 of Reviewer #2 for additional enhancements.

We enhanced the description, so that the necessary idea and concept should be understandable without reading the paper on that method.

- Benchmarks part: This section is really not clear:

o Let’s take the problem tackled in Fig3.a. I assume the authors are solving the MaxCut problem with local connection between spins where the couplings are all similar. Then, how can one define 100 different problems while this problem is unique? ie solve the MaxCut with local coupling which value is 1. Then, it follows that it is really not clear what « best run » and « average run » means in the histograms. To me, best run should be an individual run and not represented as a histogram but rather as a single data point on the plot. Same for an average run, to me it should report the average performance as a single data point on the plot.

See our response to the next point

o For Fig3.b I understand more as we can create 100 different problem instances and we have 100 data points on the plot. However, it should be written clearer in the text how many problem instances are created, how they are created and how many repetitions are performed for each problem (it is written in the text but very not clear).

We added the following subfigure to Fig. 3 to illustrate the benchmarking process:

(a) Example of solving a problem 500 times to record the best and average cut.

And extended the caption: “Benchmarks with different optimization problem sets of our OIM system (more details see “Benchmarks” in methods). Each problem set contains 100 problems, which are repeatedly solved 500 times each.”

Additionally, the figure is mentioned in the text: “As shown in Fig. 3a, we solved each problem 500 times and noted the average (red) and best (blue) solution”

- The problems used for benchmarking the chip (MaxCut) are standard problems for Ising machines but they might not be the most relevant problems for such sparse programmable connectivity graph, and it’s also not clear for which kind of problems this specific semireconfigurable connectivity would have an advantage.

o Could the authors show/ discuss the absolute advantages of their approach with respect to a standard embedding procedure? In this version, it’s not clear what is the absolute advantage of their approach with respect to other approaches used in other Ising machines (eg. Embedding or all-to-all connectivity).

The embedding was not clear in the original manuscript. Based on comments of Reviewer #2, we improved the paragraph about the embedding in the manuscript, which is copied here for convenience:

“Hence, we propose a configurable, routable network with strong local connectivity as a trade-off. The idea is, that most of the graph edges should be embedded into the local connectivity. The routing network is just used for the remaining edges, which cannot be embedded locally. While the first edges can usually be easily mapped to the hardware connectivity graph, it gets increasingly difficult to find valid mappings for the remaining edges the more nodes and edges are already mapped to a hardware component. By using the flexible routing connections on demand for these remaining challenging connections, the difficulty can be reduced. Hence, the addition of such routable channels simplifies the embedding and can increase the success rate compared to a local network [citation: 10.1109/Austrochip56145.2022.9940722].”

So, our approach uses the routable connections to make the embedding easier. But the embedding is still used. Since we unfortunately do not have a suitable, powerful embedding algorithm we cannot quantify this advantage. We do not see a meaningful comparison with an all-to-all network, since the current largest network has 48 nodes, while our sparse network has 1440 nodes. Clearly, an all-to-all network is superior to any sparse approach of the same number of nodes. However, comparing different sizes is not useful as well. At the end, it just depends on the optimization problem someone wants to solve. Sparse problems with many variables(nodes/oscillators) can be solved because sparse Ising machines tend to have more nodes. Dense (all-to-all) problems can only be solved with fewer variables (nodes/oscillators), as all-to-all OIMs are quite limited in the number of nodes.

o It is crucial to show the time for creating the embedding versus their approach on a specific problem to show a potential advantage (and if to show an advantage one has to find a specific kind of problem this is fine!). Also, the authors could report the statistics of the solutions of their approach versus the statistics of other approaches using embedding (as a claim in the introduction is to get rid of embedding that can alter the solution of a problem!). The authors could use a python package (the DWave package for instance) to simulate the embedding of a particular problem on a specific graph (Chimera, Pegasus) and the resolution with Simulated Annealing and report the results to compare with their solution and time to embed the problem.

We like the idea of such a benchmark approach. That is especially beneficial to get a performance comparison from an application point of view. However, this is a “full stack” (hardware + software) comparison, where hard+software must be competitive to yield meaningful results. Unfortunately, due to the lack of embedding algorithms, we cannot do such a “full stack” comparison.

We should note, that we do not claim to get rid of the embedding. Our idea is to make the embedding easier by providing the flexibility of the routing channels. To get rid of the embedding, an all-to-all network would be required. Presumably, the following sentence of our manuscript is mentioned by that comment: “Other approaches combine multiple physical nodes to form a single discrete variable to increase the connectivity [citations] However, this adds substantial overhead from the additionally needed physical nodes and their connections and can reduce the solution accuracy” So we just want to make the embedding more efficient.

o We want to see what the scheme/ chip really adds in value. The authors did a great job in Fig. 1a for clearly explaining the workflow for solving any problem on an Ising machine - so they could simply show numbers on the same chart of their approach versus other standard approaches (with sparse graphs + embedding).

As above: Due to the lack of a suitable embedding algorithm we cannot provide numbers for our chip. By construction of the random generated problems from the hardware connectivity graph itself, we avoided the need for embedding in our benchmarks by design.

For Fig. 1a, we used the same procedure as for the benchmarking. The problem was randomly generated from the hardware connectivity network, so that it "is already" embedded and does not need to be embedded. The solution and shown phase are actual hardware measurements.

- SAT problems are also standard problems used to benchmark Ising machines and are often sparse. My opinion is that SAT problems could be more relevant to benchmark the chip with respect to other IMs that have a sparser connectivity and rely on embedding. The authors might want to add another benchmark with SAT problems that map directly to their chip connectivity to show the real advantage of their approach.

Other works are usually using the maxcut as well. Essentially, the maxcut is equivalent to the Ising model, where just the sign of the coefficients is multiplied by -1. The maxcut is more intuitive to imagine than the coefficients of the Ising model. Hence, we think this is the best fundamental benchmark for Ising machines.

The work of Jagielski et al. has shown the transformation of the SAT to Ising machines including the software/EDA flow. At that point, it already shows the real advantage of such routable OIMs such as FPIM. We would need to copy his flow and adapt it to our system (which is very similar to his proposed FPIM) in order to solve SAT problems. We see the benefit of just adding experimental data rather small to justify this as addition to our manuscript. We think it is best to stick with the maxcut/ising problems, since they are the closest to the operation and quality of the hardware.

Minor comments:

- Page 1: "A part of bridging the gap between desired and available computing power are Ising machines »: the authors should cite the reference review paper on Ising machines here (Ising machines as hardware combinatorial solvers, Mohseni et al. Nature reviews physics, 2022)

Good point! Citation added.

- Page 1: it is written «Researchers are exploring different physical principles such as quantum mechanics³, optics^{4,5}, memristors⁶, and spintronics⁷». I understand the sentence, but while quantum mechanics could be somehow seen as a principle, optics, memristors and spintronics are not physical principles but rather physical substrats that exploit different physical principle to compute (optics with coherent Ising machines: minimum loss, memristive array leverage the Kirchhoff laws for computing, etc). Maybe the author should rephrase this sentence.

We agree, that the "physical principles" is technically incorrect. We changed this to "physical implementation":

"Researchers are exploring different physical implementations such as quantum computing⁴, optics⁵,
6, memristors⁷, and spintronics^{8–10} to directly employ ..."

- Page 2: it is written « The oscillators run and mutually manipulate their phases »: maybe the authors could find a better word than « manipulate » in this sentence (influence or modulate are the first words that come to me, but another one could be more appropriate?). Manipulate is also used 2/3 times later in the main text.

Changed to: “After the successful embedding, the oscillators run and mutually **influence** their phases, which forms a solution to the optimization problem.” and “The configurable coupler circuits convert the phase differences between the oscillators into a current, which is summed up and injected into the oscillator to **change** its phase”

- Page 2: just after, it’s not clear that the authors threshold the phase values to get spins values after measurement (while it’s clear from Figure 1). The authors should explicitly state that in the text.

Changed to: “The phases are readout to assign the discrete states of the corresponding variables oi based on their position in one of the two by π separated distinct groups.”

There is an additional section about the phase threshold added in the methods.

- Page 2: it’s not clear how the letter G was realized on the chip: only individual biases applied to the spins? No spin-spin couplings? The authors should better describe in the text (and/ or figure caption) how the letter G was realized.

The generation of the G letter is now described in the method section (“G-Letter Example Problem”).

- Page 2: "the graph embedding of the problem graph into a hardware network can be NP-hard as well » : maybe the word « connectivity graph » is to be preferred to “network ».

Good point! The usage of graph and of network is a bit inconsistent here. I want to stick with “hardware” and “problem graph” since those are clear. Change to: “Unfortunately, the graph embedding of the problem graph into a hardware connectivity network can be NP-hard as well [”

- Figure 2: a schematic of the dense local connectivity could help the reader to visualize the actual connectivity - it’s described quickly in the text (page 3) but it’s not very clear.

If I understand the comment correctly, this is already shown in Fig. 2a and mentioned in the manuscript “The connectivity of the system network is shown in Fig. 2a. Each oscillator has a local connectivity to 11 neighbors forming a fixed, sparse network. It connects to the horizontal, vertical, and diagonal neighbors...”

- Figure 3: Fig3.c is overlapping on Fig3.a (especially the x label)

Fixed! Somehow the export of the plot removed the lower part just for this subfigure.

- Page 5: "The parameters like the coupling strength were individually tuned for each problem set. » - It’s not clear how the parameters were tuned - to optimize which metric? Is it only related to Fig4.a?

Changed to: “The parameters like the coupling strength were individually tuned for all four shown benchmark problem sets to achieve the best accuracy.”

- Page 5: how can the authors sweep over the coupling value for the « weighted » problem as for this problem, each eight has a different value? (Over the available range of accessible value for the couplings)

Changed to: “Fig. 4a illustrates the impact of the coupler strength, which can be globally scaled using an external bias voltage, on the computing performances by evaluating the simple, difficult, and weighted benchmark sets at different coupling strengths”

- Can the benefit of operating at a lower frequency the oscillators be linked with the delay of propagating the spin value to distant spins? If the frequency is lower, then the period of an oscillation is longer which makes the delay less a problem?

See response to reviewer #1, (2)

In short: Yes, reducing the frequency is the “easy” way. Our research goal is to avoid that reduction in speed as the delay will add up for really large systems (or even systems of multiple dies)

- Given the locally dense connectivity the chips offer with long-range routable couplings, could the authors think of better problems than MaxCut or SAT to be solved on this IM? (I am thinking of solving more layout-driven tasks such as the convolutional neural network trained on an Ising machine (Laydevant et al., arxiv:2305.18321, 2023) but not only this specific example).

We have decided on our network without a specific application in mind. The focus is on the (analog) OIM computing and testing routable connections. The connectivity is in a similar ballpark as the Chinamere graph of D-Wave, so the work on those might be applicable to a system such as ours (Adding a few more or less couplers shouldn't cause any issues). We are not deep into the mathematical/application field, although we are planning to intensify research in this direction in the future.

The MU-MIMO problem, which is actively researched (10.1109/ICCAD57390.2023.10323680) is an application, which seems to be perfectly suitable for OIMs: Solutions need to be found fast and near optimality is likely enough.

There is some work on using the Ising model for training of AI models (like the suggested paper), but we did not yet analyze their requirements. These might be well suited for OIMs, because: A) Close to optimal solutions are likely sufficient B) Based on the neural network layers, the requirements for OIMs could be derived. Additionally, the embedding might be simplified by this regular structure. However, it is questionable if OIMs can keep up with the demand in size. Parallelization of multiple GPUs is usually done, which is much harder for OIMs.

- I have a last question, which is more a curiosity question from my side, about the Second Harmonic Injection Lockin (SHIL) used for reading out the phase of the oscillators. As the authors wrote in the manuscript, the SHIL schedule is somehow an equivalent to a temperature schedule used in simulated annealing. Can the authors comment on the choice on the SHIL schedule they have used? Have the authors tried different schedule similar to commonly used temperature schedule? ($1/\sqrt{T}$, $1/(1+\log(T))$, etc)? It's not clear which SHIL schedule has been used while it is written « The SHIM-injector gradually applies the perturbations for the SHIL... » page 4.

Compared to simulated annealing we are more restricted in terms of resolution in the hardware and only implement simple ramps. Our first generation chip was able to have up to 16 steps (4-bit resolution) and variable ramp duration. We fixed this in the second generation (to save area) to a ramping in 8 steps (3-bit), where after one nominal oscillation period (approx. 10ns) the strength is increased by one step each. (see the comment about the SHIL ramping) This seemed to deliver good performance. An unfavorable ramping can ruin the performance. We don't think the resolution is high enough to make some $1/\sqrt{t}$ or similar with these few bits. Higher resolution for the SHIL is probably not useful either, since then mismatch gets a problem. We have an external bias voltage, which controls the global strength of the injection. This could be set with a more sophisticated schedule as suggested. It would be an interesting test using slow non-linear ramps. However, ramping that bias voltage is super slow (probably multiple milliseconds) compared to the ramp on chip (approx. 100ns).

Furthermore, one can apply the SHIL, ramp it down, ramp it up again and so on, which might slightly improve the solution between iterations. This was something we could just see in simulations, but we were not able to reproduce that in our hardware (maybe noise mitigates these effects?). However, I spoke to Jaijeet Roychowdhury on the ISCAS'23 and he mentioned, that they've seen such behavior in their chip.

REVIEWERS' COMMENTS:

Reviewer #1 (Remarks to the Author):

Thank you for the revised manuscript and all my concerns have been addressed.

Reviewer #2 (Remarks to the Author):

I would like in turn to thank the authors for their very thorough answers, turning this review process into a truly enriching technical exchange.

From the Q&As, it is quite clear to me that the authors:

- have carefully considered our requests and provided thorough answers based on past published and unpublished data, that further demonstrated their advanced understanding of the current and remaining challenges associated to the design of such systems.
- have adequately modified the text in order to more accurately reflect the added value of this work.

Overall, I recommend the publication of this manuscript, with optional minor revisions (see point 3/ clarifying my remarks in the first review).

Below a follow-up on my initial main remarks.

1/ On quantifying the extent to which local connections simplify embedding: no numbers were provided, but the text modifications clarify that this is proof-of-concept work that enables tackling the engineering challenges associated to implementing non-local connections.

2/ On remarking that all-to-all connectivity is unfeasible for large graphs: this is now clearer.

3/ On benchmarking: it seems like we agree on the main points.

- I actually misinterpreted the text on comparing the chip with Gurobi, my bad.

- Regarding power vs. energy: there cannot be a perfect figure of merit unless chips are compared on equivalent problems and instances. If we could do this, then time to solution and energy would indeed be the best ones. To clarify the scope of my remark, it was mainly related to the $1\mu\text{s}/466\text{nJ}$ statement in the abstract (for example, I think Table 1 has all the necessary info), announced with no additional information on the problem being solved. If we don't know how hard the problem is, or how much more efficiently alternative hardware would perform on that same problem, then I personally would tend to convert this information into a problem-independent metrics such as power, or power per node, which at least would tell me something about how power-hungry the "building blocks" are. All I have to do then is divide $466\text{nJ}/1\mu\text{s}$, so why not do that directly, and would we lose any useful information in the process ?

- Regarding "being transparent" on problem density, I agree that the authors did provide all the necessary clues. That was poor phrasing, I meant "straightforward" rather than "transparent". For example, giving a percentage or range of non-zero terms in the adjacency matrix to give a sense of what "simple" or "difficult" means.

4/ Regarding the balance between SHIL and coupling to get the best accuracy: I personally think this is actually very interesting and would have liked to see more of that discussion in the manuscript. However I also understand that the abstract cannot be all-encompassing, and that the updated version already has enough substance for publication.

5/ Regarding the role of noise in the outcome: the new formulation is less controversial, thanks.

Reviewer #3 (Remarks to the Author):

The reviewer thanks the authors for the thorough rebuttal and detailed answers to all of the original questions/ points. The quality of the manuscript has been dramatically improved (it was already really good but the blind spots have been addressed and that makes the manuscript really enjoyable to read and makes the work more appreciable).

Thus I recommend acceptance of the manuscript and congratulate the authors for their work!

I still have two comments:

- the difficulty of the embedding procedure with routable connections might be highlighted in the discussion section

- I really think that SAT problems would be really suited for this kind of sparse graph and benchmarking on SAT would greatly benefit to the "fame" of the chip! (recent DARPA & NSF programs on Ising machines focus on SAT as benchmark problems because they find many applications in industry etc). This point is more an "advice" if I may say for the authors than an point to be addressed in the manuscript! I got (and agree to) the point of benchmarking the hardware on the proposed tasks.

Reviewer #1 (Remarks to the Author):

Thank you for the revised manuscript and all my concerns have been addressed.

Thank you for your valuable input during the review process.

Reviewer #2 (Remarks to the Author):

I would like in turn to thank the authors for their very thorough answers, turning this review process into a truly enriching technical exchange.

From the Q&As, it is quite clear to me that the authors:

- have carefully considered our requests and provided thorough answers based on past published and unpublished data, that further demonstrated their advanced understanding of the current and remaining challenges associated to the design of such systems.
- have adequately modified the text in order to more accurately reflect the added value of this work.

Overall, I recommend the publication of this manuscript, with optional minor revisions (see point 3/ clarifying my remarks in the first review).

Below a follow-up on my initial main remarks.

Thank you for your valuable input during the review process.

1/ On quantifying the extent to which local connections simplify embedding: no numbers were provided, but the text modifications clarify that this is proof-of-concept work that enables tackling the engineering challenges associated to implementing non-local connections.

We are sorry, that we can only convey our idea without providing meaningful numbers. A fast embedding with a high success rate turned out to be more difficult than initially anticipated when we started the work. We'll of course keep this in mind and will likely go more towards a co-design of the chip and its software embedding algorithms.

2/ On remarking that all-to-all connectivity is unfeasible for large graphs: this is now clearer.

Thank you very much for pointing out this blind spot in our manuscript. Your comment was more than justified, especially with the trend of "small" all-to-all networks.

3/ On benchmarking: it seems like we agree on the main points.

- I actually misinterpreted the text on comparing the chip with Gurobi, my bad.

- Regarding power vs. energy: there cannot be a perfect figure of merit unless chips are compared on equivalent problems and instances. If we could do this, then time to solution and energy would indeed be the best ones. To clarify the scope of my remark, it was mainly related to the $1\mu\text{s}/466\text{nJ}$ statement in the abstract (for example, I think Table 1 has all the necessary info), announced with no additional information on the problem being solved. If we don't know how hard the problem is, or how much more efficiently alternative hardware would perform on that same problem, then I personally would tend to convert this information into a problem-independent metrics such as power, or power per node, which at least would tell me something about how power-hungry the "building blocks" are. All I have to do then is divide $466\text{nJ}/1\mu\text{s}$, so why not do that directly, and would we lose any useful information in the process ?

Sorry, we misunderstood your remark here. We thought this was a general comment about the reporting/benchmarking methodology. Yes, your point about these two numbers in the abstract makes sense. For this, we think that power per node would be the most universal metric. Although that still lacks information about the difficulty/number of edges, it is universal to allow a comparison with different systems.

We noticed that these two numbers in the abstract (which are correct!) are a bit inconsistent with Table 1. The "less than $1\mu\text{s}$ " of the abstract was rounded up from the 950ns . The "less than 466nJ " is the highest power consumption we measured for any benchmark problem. However, the power of 460.3mW (437nJ) in Table 1 is the average among all benchmark problems. (This is affected by the actual problem, $113\mu\text{W}$ is the consumption of the oscillator and its periphery and the remaining $206\mu\text{W}$ are consumed by the couplers and a small share for the phase measurement). To avoid potential confusion from imprecise reporting, we changed this to:

"Our manufactured silicon chip, featuring 1440 oscillators implemented in a 28 nm technology, demonstrates the ability to solve optimization problems in 950ns while consuming typically $319\mu\text{W}$ per node."

- Regarding "being transparent" on problem density, I agree that the authors did provide all the necessary clues. That was poor phrasing, I meant "straightforward" rather than "transparent". For example, giving a percentage or range of non-zero terms in the adjacency matrix to give a sense of what "simple" or "difficult" means.

Sorry, we misunderstood your comment then. The number of edges for the problems are now added in the methods section, where the generation is described.

4/ Regarding the balance between SHIL and coupling to get the best accuracy: I personally think this is actually very interesting and would have liked to see more of that discussion in the manuscript. However I also understand that the abstract cannot be all-encompassing, and that the updated version already has enough substance for publication.

This is definitely a very interesting aspect, which can be even further researched to understand the internal oscillator dynamics better.

Such a discussion will touch on the internal dynamics in more detail and would require more plots. We want to keep the manuscript concise. As you pointed out, there is already a lot of content and the manuscript is getting close to the upper end of the recommended word count.

5/ Regarding the role of noise in the outcome: the new formulation is less controversial, thanks.

Thank you for pointing this out. We didn't recognize that this can be controversial when writing the manuscript. However, we can clearly see that there are multiple perspectives to interpret the influence of noise.

Reviewer #3 (Remarks to the Author):

The reviewer thanks the authors for the thorough rebuttal and detailed answers to all of the original questions/ points. The quality of the manuscript has been dramatically improved (it was already really good but the blind spots have been addressed and that makes the manuscript really enjoyable to read and makes the work more appreciable).

Thus I recommend acceptance of the manuscript and congratulate the authors for their work!

Thank you for your valuable input during the review process.

I still have two comments:

- the difficulty of the embedding procedure with routable connections might be highlighted in the discussion section

This comment is similar to 1) of reviewer #2. As this is the motivation behind the introduction of routable channels, this is mentioned in the discussion. Because we do not have meaningful quantitative data, we kept this point short on purpose.

- I really think that SAT problems would be really suited for this kind of sparse graph and benchmarking on SAT would greatly benefit to the "fame" of the chip! (recent DARPA & NSF programs on Ising machines focus on SAT as benchmark problems because they find many applications in industry etc). This point is more an "advice" if I may say for the authors than an point to be addressed in the manuscript! I got (and agree to) the point of benchmarking the hardware on the proposed tasks.

Thank you very much for this advice. We'll for sure add SAT problems to our future work and put more emphasis on those.